# LEARNING REPRESENTATIONS FOR INDEPENDENCE TESTING

## ABSTRACT

Many tools exist that attempt to detect dependence between random variables, a core question across a wide range of machine learning, statistical, and scientific endeavors. Although several statistical tests guarantee eventual detection of any dependence with enough samples, standard tests may require an exorbitant amount of samples for detecting subtle dependencies between high-dimensional random variables with complex distributions. In this work, we study two related ways to learn powerful independence tests. First, we show how to construct powerful statistical tests with finite-sample validity by using variational estimators of mutual information, such as the InfoNCE or NWJ estimators. Second, we establish a close relationship between these variational mutual information-based tests and tests based on the Hilbert-Schmidt Independence Criterion (HSIC), showing that learning a variational bound in the former case is closely related to learning kernels, typically parameterized by deep networks, in the latter. Finally, we show how to find a representation that maximizes the asymptotic power of an HSIC test, proving that this procedure works and demonstrating empirically the practical improvement of our tests (with HSIC tests generally outperforming the variational ones) on difficult problems of detecting structured dependence.

## 1 INTRODUCTION

Independence testing, the question of using paired samples to determine whether a random variable $X$ and another $Y$ are associated with one another or if they are statistically independent, is one of the most common tasks across scientific and data-based fields. Traditional methods make strong parametric assumptions, for instance assuming that $X$ and $Y$ are jointly normal so that dependence is characterized by covariance, and/or operate only in limited settings, for instance the tabular setting of the celebrated $\chi^2$ test or Fisher's exact test. Applying these approaches to high-dimensional continuous data is difficult at best.

One characterization of independence is the (Shannon) mutual information (MI): this quantity is zero if two random variables are independent, and positive if they are dependent. Substantial effort has been made in estimating this quantity with a variety of estimators; see e.g. the broad list based on binning, nearest-neighbors, kernel density estimation, and so on implemented by Szabó (2014). In high dimensions, however, recent work has focused on estimating variational bounds defined by deep networks (see e.g. Poole et al., 2019), which can ideally learn problem-specific structure. To our knowledge, this class of estimators has not yet been used to construct *statistical tests* of independence: in particular, if we run the estimation algorithm and estimate a lower bound on the MI of $0.12$, does that mean that the variables are (perhaps weakly) dependent, or might that be due to random noise on the samples we saw? The first contribution of this paper is to construct and evaluate tests of this form.

Mutual information, though, is notoriously difficult to estimate (Paninski, 2003). If the question we want to ask is "are $X$ and $Y$ dependent," we can also consider turning to a different characterization of dependence which may be statistically easier to estimate. The Hilbert-Schmidt Independence Criterion (HSIC), introduced by Gretton et al. (2005), measures the total cross-covariance between feature representations of $X$ and $Y$, and can be estimated from samples efficiently. The construction supports even *infinite-dimensional* features in a reproducing kernel Hilbert space (RKHS), equivalent to choosing a kernel function. With appropriate choices of kernel (see Szabó & Sriperumbudur, 2018), the HSIC is also zero if and only if $X \perp\!\!\!\perp Y$. Székely et al. (2007) and Lyons (2013) separately

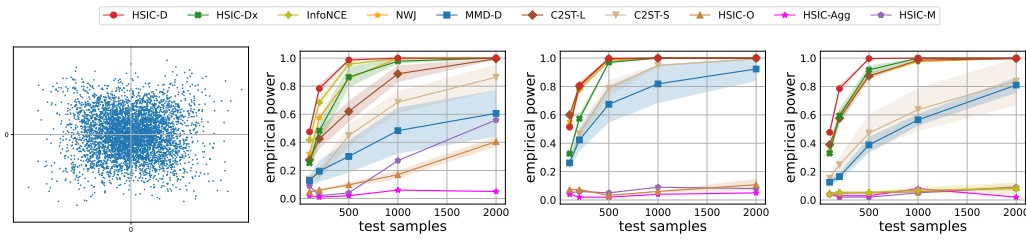

(a) Samples $X, Y \sim \mathbb{P}_{xy}$  (b) Power vs. $m$; $d = 4$  (c) Power vs. $m$; $d = 10$  (d) Power vs. $m$; $d = 15$

Figure 1: Vanilla kernel-based HSIC tests struggle on high-dimensional data. (a) A bimodal Gaussian mixture, with visible dependence between $X$ and $Y$. (b)-(d) Test power as we add an increasing number of independent noise dimensions. When $d = 4$ the median heuristic (HSIC-M) confidently detects dependence between $X$ and $Y$ with a reasonably large sample size, but struggles when $d = 10$ or $d = 15$. Our method (HSIC-D) detects dependence in high dimensions with many fewer samples.

proposed *distance covariance* tests, which measure the covariance of pairwise distances between $X$ values with distances between $Y$ values; this can be viewed as HSIC with a particular kernel (Sejdinovic et al., 2013). Our second contribution describes how HSIC, in fact, is a lower bound on mutual information, and that tests based on either are closely related.

Alternatively, we can characterize an independence problem as a two-sample one. Two-sample problems are concerned with the question: given samples from $\mathbb{P}$ and $\mathbb{Q}$, is $\mathbb{P} = \mathbb{Q}$? Under this framework, when we consider samples from the joint distributions $\mathbb{P}_{xy}$ and $\mathbb{P}_x \times \mathbb{P}_y$, the two-sample problem characterizes an equivalent independence problem between variables $X$ and $Y$. One way we can measure the discrepancy between $\mathbb{P}_{xy}$ and $\mathbb{P}_x \times \mathbb{P}_y$ is with a class of distances on probability measures called integral probability metrics (IPM). Defining an IPM using critic functions in an RKHS, we recover the kernel-based Maximum-Mean Discrepancy (MMD) Gretton et al. (2012a).

Any reasonable choice of kernel, such as the Gaussian kernel with unit bandwidth or the distance kernel that recovers distance covariance, will *eventually* be able to detect any fixed dependence with enough samples. In practice, however, this scheme can perform extremely poorly; if the data varies on a very different scale, it will take exorbitant quantities of data to achieve any reasonable test power.

Thus, tests using Gaussian kernels often rely on the *median heuristic* to choose a kernel relevant to the data at hand: choosing a bandwidth based on the median pairwise distance among data points (Gretton et al., 2012a). While this is a reasonable first guess for many data types, there exist datasets where it can be dramatically better to instead select a bandwidth that optimizes a measure of test power (Sutherland et al., 2017). Beyond that, there exist many distributions where no Gaussian kernel performs well – for instance, many problems on natural image data or involving sparsity – but Gaussian kernels applied to latent representations of such data do. As an extreme example, consider random variables whose twelfth through fifteenth decimal places are always equal. Representations based on Euclidean distance will require an exorbitant number of samples to detect dependence, but a representation that extracts only the relevant decimal places will do so very quickly.

This concept similarly applies to tests based on mutual information. In fact, we can view estimation of a variational MI bound as learning a representation of the data that maximizes its measure of dependency. Thus, learning representations of the data that make any dependency more explicit is central to developing more powerful independence tests. Our final contribution involves developing a scheme to learn these optimal representations for both kernel-based and certain MI-based independence tests.

Our kernel-based method method involves learning the representation that maximizes the asymptotic power of a test based on HSIC. This expression is dominated by the signal-to-noise ratio of HSIC to its standard deviation. We show that we can estimate this quantity from finite samples in quadratic time, and that it is consistent. We then construct a valid test via data splitting and permutation testing (e.g. Rindt et al., 2021). Our overall approach builds on prior work in kernel two-sample testing (Gretton et al., 2012b; Jitkrittum et al., 2016; Sutherland et al., 2017; Liu et al., 2020), but translated to the independence testing setting. Our comprehensive experiments also provide empirical evidence that a reframing of the MMD-based two-sample test into an independence test is not optimal; a phenomenon we attribute to the statistical properties of both estimators.

## 2  TESTS BASED ON VARIATIONAL MUTUAL INFORMATION BOUNDS

**Problem 1** (Independence testing). *Let $Z = (X, Y) \sim \mathbb{P}_{xy}$ on a domain $\mathcal{X} \times \mathcal{Y}$, and $\mathbb{P}_x$ and $\mathbb{P}_y$ the corresponding marginal distributions on $\mathcal{X}$ and $\mathcal{Y}$. We observe $m$ independent samples $\{(x_1, y_1), \ldots, (x_m, y_m)\}$ from $\mathbb{P}_{xy}$. We conduct a null hypothesis significance test, with the null hypothesis $\mathfrak{H}_0 : \mathbb{P}_{xy} = \mathbb{P}_x \times \mathbb{P}_y$ (so $X \perp\!\!\!\perp Y$), and the alternative $\mathfrak{H}_1 : \mathbb{P}_{xy} \neq \mathbb{P}_x \times \mathbb{P}_y$ (i.e. $X \not\perp\!\!\!\perp Y$).*

We wish to solve this problem without making strong (parametric) assumptions about the form of $\mathbb{P}_{xy}$, $\mathbb{P}_x$, or $\mathbb{P}_y$. Most independence tests are based on estimating the "amount" of dependence between $X$ and $Y$, or equivalently the discrepancy between between $\mathbb{P}_{xy}$ and $\mathbb{P}_x \times \mathbb{P}_y$. Given a nonnegative quantity which is zero if $X \perp\!\!\!\perp Y$, we can reject $\mathfrak{H}_0$ if our estimate is large enough that we are confident the true value is positive.

The most famous such quantity is the (Shannon) mutual information. It can be defined as the Kullback-Liebler divergence of $\mathbb{P}_{xy}$ from $\mathbb{P}_x \times \mathbb{P}_y$, and is zero if and only if $X \perp\!\!\!\perp Y$; equivalently, it is the amount by which knowledge of $Y$ decreases the entropy of $X$. While many estimators exist based roughly on various forms of density estimation (see e.g. Szabó, 2014), as discussed above these can fail to detect subtle or structured forms of dependence with reasonable numbers of samples. Variational estimators of mutual information give an opportunity for problem-specific representations. As an example, consider the following lower bound (van den Oord et al., 2018; Poole et al., 2019), called InfoNCE for its connection to noise-contrastive estimation (Gutmann & Hyvärinen, 2012):

$$\mathrm{I}_{\mathrm{NCE}}^{f,K}(X;Y) = \mathbb{E}_{(x_i, y_i) \sim \mathbb{P}_{xy}^K} \left[ \hat{\mathrm{I}}_{\mathrm{NCE}}^{f,K} \right] \leq \mathrm{I}(X;Y) \qquad \hat{\mathrm{I}}_{\mathrm{NCE}}^{f,K} = \frac{1}{K} \sum_{i=1}^{K} \log \frac{e^{f(x_i, y_i)}}{\frac{1}{K} \sum_{j=1}^{K} e^{f(x_i, y_j)}}.$$

Here $K$ is a batch size, which in our setting will typically be the total number of available points. Each $f : \mathcal{X} \times \mathcal{Y} \to \mathbb{R}$ leads to a different lower bound; the largest $\mathrm{I}_{\mathrm{NCE}}^{f,K}$ is the tightest bound. In practice, users typically parameterize $f$ as a deep network and maximize $\hat{\mathrm{I}}_{\mathrm{NCE}}^{f,K}$ on minibatches from a training set, providing an opportunity for $f$ to learn useful feature adapted to the problem at hand. There are a variety of bounds of this general type; Poole et al. (2019) give a unified accounting of many. Different bounds yield different bias-variance tradeoffs, but run up against various limitations on the possibility of estimation (Song & Ermon, 2020; McAllester & Stratos, 2020).

For independence testing, the key question is whether the true value $\mathrm{I}(X;Y) > 0$. This is guaranteed if a lower bound, such as $\mathrm{I}_{\mathrm{NCE}}^{f,K}$ for some particular $f$ and $K$, is positive *at the population level*, i.e. with the true expectation. To estimate this, by far the easiest approach is data splitting: choose $f$ to maximize $\hat{\mathrm{I}}_{\mathrm{NCE}}^{f,K}$ on (minibatches from) a training set, then evaluate $\hat{\mathrm{I}}_{\mathrm{NCE}}^{f,K}$ on the heldout test set.

How large should $\hat{\mathrm{I}}_{\mathrm{NCE}}^{f,K}$ be in order to be confident that $\mathrm{I}_{\mathrm{NCE}}^{f,K} > 0$? That is, what does the distribution of $\hat{\mathrm{I}}_{\mathrm{NCE}}^{f,K}$ look like when $\mathrm{I}(X;Y) = 0$, and hence $\mathrm{I}_{\mathrm{NCE}}^{f,K} \leq 0$? For a given $f$, we can answer this question with *permutation testing*, which estimates values under the null hypothesis ($\mathbb{P}_{xy} = \mathbb{P}_x \times \mathbb{P}_y$) by randomly shuffling the test data, breaking dependence. To construct a test with probability of false rejection at most $\alpha$, we can compute the empirical $(1 - \alpha)$ quantile from this permuted set, as long as we include the original paired data in this shuffling (Hemerik & Goeman, 2018, Theorem 2). We reject the null hypothesis if this quantile is smaller than the test statistic

$$\hat{\mathrm{I}}_{\mathrm{NCE}}^{f,K} = \frac{1}{K} \sum_{i=1}^{K} f(x_i, y_i) - \frac{1}{K} \sum_{i=1}^{K} \log \left( \frac{1}{K} \sum_{j=1}^{K} e^{f(x_i, y_j)} \right). \tag{1}$$

Written in this form, notice that the second term of $\hat{\mathrm{I}}_{\mathrm{NCE}}$ is permutation-invariant: the test statistic and each of its permuted versions are $\frac{1}{K} \sum_{i=1}^{K} f(x_i, y_i)$ shifted by the same constant. Thus, although this second term plays a vital role in selecting the critic function $f$, *at test time* the only thing that matters is whether the mean value of $f(x, y)$ is higher for the true pairings than for random pairs. The same is true for the NWJ (Nguyen et al., 2010), DV (Donsker & Varadhan, 1983), $\mathrm{I}_{\mathrm{JS}}$, and $\mathrm{I}_\alpha$ lower bounds as discussed by Poole et al. (2019), as well as the MINE estimator (Belghazi et al., 2018).

## 3 TESTS WITH THE HILBERT-SCHMIDT INDEPENDENCE CRITERION (HSIC)

The Hilbert-Schmidt Independence Criterion (HSIC, Gretton et al., 2005) is also zero if and only if $X$ and $Y$ are independent when an appropriate kernel is chosen. Unlike the mutual information, it is easy to estimate from samples. We will expand on its relationship to mutual information in Section 4.

To define HSIC, we first briefly review positive-definite kernels. A (real-valued) *kernel* is a function $k : \mathcal{X} \times \mathcal{X} \to \mathbb{R}$ which can be expressed as the inner product between feature maps $\phi : \mathcal{X} \to \mathcal{F}$, $k(x, x') = \langle \phi(x), \phi(x') \rangle_\mathcal{F}$, where $\mathcal{F}$ is any Hilbert space. A special case is $\mathcal{F} = \mathbb{R}^p$ and $k(x, x') = \phi(x) \cdot \phi(x')$. For every kernel function, there exists a unique space called the *reproducing kernel Hilbert space* (RKHS), which consists of functions $f : \mathcal{X} \to \mathbb{R}$. The key *reproducing property* of an RKHS $\mathcal{F}$ states that for any function $f \in \mathcal{F}$ and any point $x \in \mathcal{X}$, we have $\langle f, \phi(x) \rangle_\mathcal{F} = f(x)$.

Suppose we have a kernel $k$ on $\mathcal{X}$ with RKHS $\mathcal{F}$ and feature map $\phi$, as well as another kernel $l$ on $\mathcal{Y}$ with RKHS $\mathcal{G}$ and feature map $\psi$. Let $\otimes$ denote the outer product.[1] The *cross-covariance operator* is

$$C_{xy} = \mathop{\mathbb{E}}_{xy} \Big[ \big( \phi(x) - \mathop{\mathbb{E}}_x \phi(x) \big) \otimes \big( \psi(y) - \mathop{\mathbb{E}}_y \psi(y) \big) \Big];$$

for kernels with finite-dimensional feature maps, this is exactly the standard (cross-)covariance matrix between the features of $X$ and those of $Y$. Under mild integrability conditions on the kernel and the distributions,[2] the reproducing property shows that $\langle f, C_{xy} g \rangle = \mathrm{Cov}(f(X), g(Y))$ for all $f \in \mathcal{F}$, $g \in \mathcal{G}$. One definition of independence is whether there exist any correlated "test functions" $f$ and $g$. Thus, for rich enough choices of kernel – using universal $k$ and $l$ suffices, but is not necessary (Szabó & Sriperumbudur, 2018) – we have that $X \perp\!\!\!\perp Y$ if and only if the operator $C_{xy} = 0$. We can thus check whether the operator is zero, and hence whether $X \perp\!\!\!\perp Y$, by checking the squared Hilbert-Schmidt norm of $C_{xy}$, $\mathrm{HSIC}(X, Y) = \|C_{xy}\|_{\mathrm{HS}}^2$. With finite-dimensional features, this corresponds to the squared Frobenius norm of the feature cross-covariance matrix.

Another way to interpret HSIC is as a distance between $\mathbb{P}_{xy}$ and $\mathbb{P}_x \times \mathbb{P}_y$, similarly to how the mutual information is the KL divergence between those same distributions.

**Proposition 2** (Gretton et al., 2012a, Theorem 25). *Let $k$ and $l$ be kernels on $\mathcal{X}$ and $\mathcal{Y}$, and define a kernel on $\mathcal{X} \times \mathcal{Y}$ by $h\big((x, y), (x', y')\big) = k(x, x') l(y, y')$ with RKHS $\mathcal{H}$. Then*

$$\sqrt{\mathrm{HSIC}_{k,l}(X, Y)} = \mathrm{MMD}_h(\mathbb{P}_{xy}, \mathbb{P}_x \times \mathbb{P}_y) = \sup_{\substack{f \in \mathcal{H} \\ \|f\|_\mathcal{H} \leq 1}} \mathop{\mathbb{E}}_{(X,Y) \sim \mathbb{P}_{xy}} [f(X, Y)] - \mathop{\mathbb{E}}_{\substack{X \sim \mathbb{P}_x \\ Y' \sim \mathbb{P}_y}} [f(X, Y')]$$

$$= \sqrt{\mathop{\mathbb{E}}_{\substack{(X,Y),(X',Y') \sim \mathbb{P}_{xy} \\ Y'',Y''' \sim \mathbb{P}_y}} \Big[ k(X, X') l(Y, Y') - 2 k(X, X') l(Y, Y'') + k(X, X') l(Y'', Y''') \Big]}.$$

Taking the last form and rearranging to save repeated computation yields two similar, popular estimators of HSIC. "The biased estimator" is $\mathrm{HSIC}(\hat{P}_{xy}, \hat{P}_x \times \hat{P}_y)$ for empirical distributions $\hat{P}$:

$$\widehat{\mathrm{HSIC}}_\mathrm{b}(X, Y) = \frac{1}{m^2} \langle \mathbf{K}, \mathbf{HLH} \rangle_F \qquad \text{for } \langle A, B \rangle_F = \sum_{ij} A_{ij} B_{ij}, \tag{2}$$

where here $\mathbf{K}$ is the $m \times m$ matrix with entries $k_{ij} = k(x_i, x_j)$, $\mathbf{L}$ similarly has entries $l_{ij} = l(y_i, y_j)$, and $\mathbf{H}$ is the "centering matrix" $\mathbf{I}_m - \frac{1}{m} \mathbf{1}_m \mathbf{1}_m^\top$ (so the estimator can be easily implemented without matrix multiplication). This estimator has $\mathcal{O}(1/m)$ bias, but is consistent.

The other common estimator, "the unbiased estimator," is a $U$-statistic (Song et al., 2012):

$$\widehat{\mathrm{HSIC}}_\mathrm{u}(X, Y) = \frac{1}{m(m-3)} \left[ \langle \tilde{K}, \tilde{L} \rangle_F + \frac{\mathbf{1}_m^\top \tilde{K} \mathbf{1}_m \mathbf{1}_m^\top \tilde{L} \mathbf{1}_m}{(m-1)(m-2)} - \frac{2 \mathbf{1}_m^\top \tilde{K} \tilde{L} \mathbf{1}_m}{m-2} \right], \tag{3}$$

where $\tilde{K}$ and $\tilde{L}$ are $m \times m$ matrices whose diagonal entries are zero but whose off-diagonal entries agree with those of $\mathbf{K}$ or $\mathbf{L}$. It is unbiased, $\mathbb{E} \, \widehat{\mathrm{HSIC}}_\mathrm{u}(X, Y) = \mathrm{HSIC}(X, Y)$; it is also consistent, and can be computed in the same $\mathcal{O}(m^2)$ time as $\widehat{\mathrm{HSIC}}_\mathrm{b}$, without matrix multiplication.

---

[1] Analogously to Euclidean outer products, $f \otimes g : \mathcal{G} \to \mathcal{F}$ is given by $[f \otimes g] g' = f \langle g, g' \rangle_\mathcal{G}$.

[2] It suffices that $\mathbb{E}[\sqrt{k(x, x) l(y, y)}] < \infty$; this is guaranteed for any distributions when $k, l$ are bounded.

Either statistic can be used with a permutation test to construct a test of independence with finite-sample validity, in the same way as described for mutual information bounds. Unlike in that case, by simply picking e.g. distance or Gaussian kernels for $k$ and $l$, we can guarantee that if $X \not\perp\!\!\!\perp Y$ then $\mathrm{HSIC}(X, Y) > 0$, in which case $\sqrt{m}(\widehat{\mathrm{HSIC}}_{\mathrm{u}} - \mathrm{HSIC}) \xrightarrow{\mathrm{d}} \mathcal{N}(0, \sigma_{\mathfrak{H}_1}^2)$ (Song et al., 2012, Theorem 5). By contrast, when $\mathrm{HSIC} = 0$, it is $m\,\widehat{\mathrm{HSIC}}_{\mathrm{u}}$ (rather than $\sqrt{m}\widehat{\mathrm{HSIC}}_{\mathrm{u}}$) that converges in distribution to something with complex dependence on $\mathbb{P}_x$, $\mathbb{P}_y$, $k$, and $l$. Thus, if we consider a test statistic $m\,\widehat{\mathrm{HSIC}}_{\mathrm{u}}$, when $\mathrm{HSIC} = 0$ this statistic has mean zero and standard deviation $\Theta(1)$. When $\mathrm{HSIC} > 0$, though, the statistic has mean and standard deviation $\Theta(\sqrt{m}) \to \infty$. Thus, as $m \to \infty$, eventually the test will reject if $X \not\perp\!\!\!\perp Y$ (Rindt et al., 2021).

## 4    CONNECTING HSIC AND MUTUAL INFORMATION TESTS

**HSIC as an MI lower bound.**    Suppose the kernel $h$ on $(x, y)$ pairs of Proposition 2 is bounded: $\sup_{x \in \mathcal{X}, y \in \mathcal{Y}} h((x, y), (x, y)) \leq \nu^2$ for $\nu \geq 0$. Then $|f(x, y)| = |\langle f, \phi_h(x, y)\rangle_{\mathcal{H}}| \leq \nu\|f\|_{\mathcal{H}}$ by the reproducing property and Cauchy-Schwarz, so $\|f\|_{\mathcal{H}} \geq \frac{1}{\nu}\sup_{x,y}|f(x, y)| = \frac{1}{\nu}\|f\|_{\infty}$. We thus have

$$\sqrt{\mathrm{HSIC}(X, Y)} \leq \sup_{f: \|f\|_{\infty} \leq \nu} \mathbb{E}_{(X,Y) \sim \mathbb{P}_{xy}}[f(X, Y)] - \mathbb{E}_{\substack{X \sim \mathbb{P}_x \\ Y' \sim \mathbb{P}_y}}[f(X, Y')] = 2\nu\,\mathrm{TV}(\mathbb{P}_{xy}, \mathbb{P}_x \times \mathbb{P}_y), \quad (4)$$

where TV is the total variation distance between distributions (Sriperumbudur et al., 2012)[3] Applying standard bounds relating the total variation to the KL divergence, we obtain the following.

**Proposition 3.** *In the setting of Proposition 2, suppose $\sup_{x \in \mathcal{X}, y \in \mathcal{Y}} h((x, y), (x, y)) \leq \nu^2$. Then*

$$\frac{1}{2\nu^2}\mathrm{HSIC}(X, Y) \leq \mathrm{I}(X; Y) \qquad and \qquad -\log\left(1 - \frac{1}{4\nu^2}\mathrm{HSIC}(X, Y)\right) \leq \mathrm{I}(X; Y).$$

*Proof.* The first bound applies Pinsker's inequality, which relates total variation to KL, to (4). The second instead applies the bound of Bretagnolle & Huber (1978) (also see Canonne, 2023). $\qquad\square$

The second bound is tighter for large values of $\mathrm{I}(X; Y)$, but both are monotonic in $\mathrm{HSIC}(X, Y)/\nu^2$. We could thus consider, as in Section 2, choosing kernels $k$, $l$ to maximize a lower bound on $\mathrm{I}(X; Y)$, by maximizing $\mathrm{HSIC}(X, Y)/\nu^2$. Indeed, maximizing HSIC has been used by previous applications in many areas (e.g. Blaschko & Gretton, 2009; Song et al., 2012; Li et al., 2021; Dong et al., 2023).

**HSIC and variational MI tests.**    Now consider a test based on $\mathrm{MMD}_f(\mathbb{P}_{xy}, \mathbb{P}_x \times \mathbb{P}_y)$ using a kernel of the form $h((x, y), (x', y')) = f(x, y)f(x', y')$ for some real-valued function $f$. (If $f(x, y) = f_1(x)f_2(y)$, this is an HSIC test.) Because $\phi_h(x, y) = f(x, y) \in \mathbb{R}$ is a valid feature map, every function in $\mathcal{H}$ is of the form $\alpha f(x, y)$ with $\|\alpha f\|_{\mathcal{H}} = |\alpha|$. By Proposition 2,

$$\mathrm{MMD}_f(\mathbb{P}_{xy}, \mathbb{P}_x \times \mathbb{P}_y)^2 = \left(\sup_{|\alpha| \leq 1} \alpha\left(\mathbb{E}\,f(X, Y) - \mathbb{E}\,f(X, Y')\right)\right)^2 = \left(\mathbb{E}\,f(X, Y) - \mathbb{E}\,f(X, Y')\right)^2.$$

The plug-in estimator corresponding to the biased HSIC estimator (2) would yield the test statistic

$$\widehat{\mathrm{MMD}}_{\mathrm{b}}^2 = \left(\frac{1}{m}\sum_{i=1}^{m} f(x_i, y_i) - \frac{1}{m^2}\sum_{i=1}^{m}\sum_{j=1}^{m} f(x_i, y_j)\right)^2. \quad (5)$$

(If $f(x, y) = f_1(x)f_2(y)$, this is $\widehat{\mathrm{HSIC}}_{\mathrm{b}}$.) Comparing (5) to (1) with $K = m$, we can see that the main term $\hat{T} = \frac{1}{m}\sum_i f(x_i, y_i)$ is identical. The other term is permutation-invariant; it is the mean of $\hat{T}$ over all possible permutations, $\bar{T}$. Thus, a permutation test based on (5) asks how far the value of $\hat{T}$ for the true data is from $\bar{T}$, while a test based on (1) is equivalent to asking how much the value of $\hat{T}$ exceeds $\bar{T}$. The only difference is that (5) gives a two-sided test, while (1) is a one-sided test.[4]

---

[3]Sriperumbudur et al. (2012) define the TV as twice the more common definition, which we use here.

[4]Li et al. (2021, Section 3.1) also found a relationship between HSIC and $\mathrm{I}_{\mathrm{NCE}}$ for categorical $Y$.

Our usual test uses $\widehat{\mathrm{HSIC}}_{\mathrm{u}}$ of (3) instead of $\widehat{\mathrm{HSIC}}_{\mathrm{b}}$, but the difference in estimators is typically small. Thus, if we use deep kernels of the form $k(x, x') = f(x)f(x')$ and $l(y, y') = g(y)g(y')$, the HSIC test[5] is nearly equivalent to the NCE test with a separable critic function $(x, y) \mapsto f(x)g(y)$. The NCE test chooses a critic by maximizing the NCE estimate; while we could choose a (different) critic by maximizing the HSIC, we will now see there is a better approach.

## 5 LEARNING REPRESENTATIONS FOR INDEPENDENCE TESTING

Typically, a mutual information lower bound is considered better if the population value of the bound is larger: that makes it a tighter bound. This viewpoint, however, neglects the issue of statistical estimation of that bound, e.g. the difference between $\mathrm{I}_{\mathrm{NCE}}^{f,K}$ and $\hat{\mathrm{I}}_{\mathrm{NCE}}^{f,K}$. The best statistical test, among tests with appropriate Type I (false rejection) control, is the one with highest *test power*: the probability of correctly rejecting the null hypothesis $\mathfrak{H}_0$ when $X \not\perp\!\!\!\perp Y$.

To warm up, we can first consider the behavior of a permutation test based on (1) or the many other mutual information bounds based only on $\hat{T} = \frac{1}{m} \sum_{i=1}^{m} f(x_i, y_i)$. Each bound will choose a different $f$ during training, but at test time, only $\hat{T}$ matters. Let $T = \mathbb{E}_{\mathbb{P}_{xy}} f(x, y)$ and $T_0 = \mathbb{E}_{\mathbb{P}_x \times \mathbb{P}_y} f(x, y)$. Assuming $0 < \sigma^2 = \mathrm{Var}\, f(x, y) < \infty$, the central limit theorem implies that $\frac{1}{\sigma}\sqrt{m}(\hat{T} - T) \xrightarrow{\mathrm{d}} \mathcal{N}(0, 1)$. Then the rejection threshold $r$ should be such that, denoting $a \sim b$ to mean $\lim_{m\to\infty} a/b = 1$, and using $\Phi$ as the standard normal cdf,

$$\alpha = \Pr_{\mathfrak{H}_0}(\hat{T} > r) = \Pr_{\mathfrak{H}_0}\left( \frac{\hat{T} - T_0}{\sigma_{\mathfrak{H}_0}/\sqrt{m}} > \frac{r - T_0}{\sigma_{\mathfrak{H}_0}/\sqrt{m}} \right) \sim \Phi\left( \frac{T_0 - r}{\sigma_{\mathfrak{H}_0}/\sqrt{m}} \right);$$

implying $r \sim T_0 - \Phi^{-1}(\alpha)\frac{\sigma_{\mathfrak{H}_0}}{\sqrt{m}}$. Similar logic tells us that the finite sample power is roughly (recalling $T, T_0, \sigma_{\mathfrak{H}_0}, \sigma_{\mathfrak{H}_1}$ do not depend on $m$)

$$\Pr_{\mathfrak{H}_1}\left(\hat{T} > r\right) \sim \Phi\left( \frac{T - r}{\sigma_{\mathfrak{H}_1}/\sqrt{m}} \right) \sim \Phi\left( \sqrt{m}\, \frac{T - T_0}{\sigma_{\mathfrak{H}_1}} + \frac{\sigma_{\mathfrak{H}_0}}{\sigma_{\mathfrak{H}_1}} \Phi^{-1}(\alpha) \right).$$

This argument shows that, as long as $T > T_0$ and $\hat{T}$ has finite positive variance under both $\mathfrak{H}_0$ and $\mathfrak{H}_1$, a test based on $\hat{T}$ will eventually reject any fixed alternative with probability 1. How quickly in $m$ it reaches high power, however, is dominated by the signal-to-noise ratio (SNR) $(T - T_0)/\sigma_{\mathfrak{H}_1}$. When we choose a test based on maximizing the value of a mutual information lower bound, we maximize a criterion such as (1), which does not directly correspond to the test power. Instead, we would perhaps be better served by maximizing $(T - T_0)/\sigma_{\mathfrak{H}_1}$, or rather, an empirical estimate of that quantity. We will shortly see that a very similar argument applies to HSIC as well, with essentially the same signal-to-noise ratio.

**Asymptotic test power (HSIC).** Similarly to the aforementioned $\hat{T}$-based tests, we can analyze the power of tests based on HSIC. Letting $\Phi$ denote the standard normal CDF and $r$ be the threshold found by permutation testing, Proposition 5 implies the asymptotic test power is

$$\Pr_{\mathfrak{H}_1}\left( m\,\widehat{\mathrm{HSIC}}_{\mathrm{u}} > r \right) = \Pr_{\mathfrak{H}_1}\left( \frac{\sqrt{m}}{\sigma_{\mathfrak{H}_1}}(\widehat{\mathrm{HSIC}}_{\mathrm{u}} - \mathrm{HSIC}) > \frac{r}{\sqrt{m}\,\sigma_{\mathfrak{H}_1}} - \frac{\sqrt{m}\,\mathrm{HSIC}}{\sigma_{\mathfrak{H}_1}} \right)$$

$$\sim \Phi\left( \frac{\sqrt{m}\,\mathrm{HSIC}}{\sigma_{\mathfrak{H}_1}} - \frac{r}{\sqrt{m}\,\sigma_{\mathfrak{H}_1}} \right), \tag{6}$$

where $a \sim b$ means $\lim_{m\to\infty} a/b = 1$. It follows that for any given $\mathbb{P}_{xy}$, $k$, and $l$, HSIC and $\sigma_{\mathfrak{H}_1}$ are constants, and Proposition 5 tells us that $r$ also converges to a constant. Thus, as the sample size $m$ grows, the test power is dominated by the first term[6], and so maximizing the following *approximate*

---

[5]This version is closely related to a witness two-sample test (Kübler et al., 2022) used for independence.

[6]This statement, while true in the limit $m \to \infty$, does not fully describe the setting for finite $m$: for instance, a test which is only near the rejection threshold will have $r \approx m\,\mathrm{HSIC}$, in which case the two terms are of approximately equal size. As in Footnote 10, though, we can estimate the power of an $m$-sample test with $n$ samples: here we use $n$ samples to approximately maximize the power of a test with $m \to \infty$ samples. Deka & Sutherland (2023) estimate an analogue of (6) rather than just (7) in a related setting.

*test power* maximizes the limiting power of the test as $m \to \infty$:

$$J(X, Y; k, l) = \frac{\text{HSIC}(X, Y; k, l)}{\sigma_{\mathfrak{H}_1}(X, Y; k, l)}. \tag{7}$$

This signal-to-noise ratio is parallel to the one for $\hat{T}$-based tests. Indeed, as argued in Section 4, when considering kernels $k(x, x') = f_1(x)f_1(x')$ and $l(y, y') = f_2(y)f_2(y')$, an estimate of $J$ based on the biased HSIC estimator yields the exact same SNR as for $\hat{T}$-based tests when $f(x, y) = f_1(x)f_2(y)$. In practice, we choose to estimate HSIC with its unbiased estimator so that

$$\hat{J}_\lambda(X, Y; k, l) = \frac{\widehat{\text{HSIC}}_u(X, Y; k, l)}{\hat{\sigma}^\lambda_{\mathfrak{H}_1}(X, Y; k, l)}, \tag{8}$$

where $\hat{\sigma}^\lambda_{\mathfrak{H}_1}(X, Y; k) = \sqrt{16(R - \widehat{\text{HSIC}}_u^2) + \lambda}$ is a regularized variance estimator. Here $R$ is exactly the estimate from Proposition 5; following Song et al. (2012), it can be computed more efficiently with $R = \frac{((n-4)!)^2}{4n((n-1)!)^2}\|\boldsymbol{h}\|^2$, where the vector $\boldsymbol{h}$ is

$$\boldsymbol{h} = (n-2)^2 \left(\tilde{\boldsymbol{K}} \circ \tilde{\boldsymbol{L}}\right)\mathbf{1} - n(\tilde{\boldsymbol{K}}\mathbf{1}) \circ (\tilde{\boldsymbol{L}}\mathbf{1}) + (\mathbf{1}^\top\tilde{\boldsymbol{L}}\mathbf{1})\tilde{\boldsymbol{K}}\mathbf{1} + (\mathbf{1}^\top\tilde{\boldsymbol{K}}\mathbf{1})\tilde{\boldsymbol{L}}\mathbf{1} - (\mathbf{1}^\top\tilde{\boldsymbol{K}}\tilde{\boldsymbol{L}}\mathbf{1})\mathbf{1}$$
$$+ (n-2)\left((\mathbf{1}^\top(\tilde{\boldsymbol{K}} \circ \tilde{\boldsymbol{L}})\mathbf{1})\mathbf{1} - \tilde{\boldsymbol{K}}\tilde{\boldsymbol{L}}\mathbf{1} - \tilde{\boldsymbol{L}}\tilde{\boldsymbol{K}}\mathbf{1}\right),$$

with $\circ$ denoting elementwise multiplication on matrices, and $\mathbf{1} = (1, \ldots, 1) \in \mathbb{R}^n$. Given sets of possible kernels, we can approximately identify the kernels $(k, l)$ yielding the asymptotically most powerful HSIC test by maximizing (8). Typically, we run some variant of a gradient-based optimization algorithm to maximize (8) with respect to the parameters of $k$ and/or $l$.

**Kernel architecture.** One way we can perform an HSIC-based test on representations of the data is by incorporating a featurizer that acts on samples just before passing through a kernel. When our feature mapping is parameterized by deep networks, we get the class of deep kernels (Wilson et al., 2016), which have been successfully used in two-sample testing (Sutherland et al., 2017; Liu et al., 2020; 2021) and many other settings (e.g. Li et al., 2017; Arbel et al., 2018; Jean et al., 2018; Li et al., 2021). We use the following deep kernels for $X$ and $Y$:

$$k_\omega(x, x') = (1 - \epsilon_X)\,\kappa_X(f_\omega(x), f_\omega(x')) + \epsilon_X\,q_X(x, x')$$
$$l_\gamma(y, y') = (1 - \epsilon_Y)\,\kappa_Y(g_\gamma(y), g_\gamma(y')) + \epsilon_Y\,q_Y(y, y').$$

Here $f_\omega$ and $g_\gamma$ are deep networks with parameters in $\omega, \gamma$, which extract relevant features from $\mathcal{X}$ or $\mathcal{Y}$ to a feature space $\mathbb{R}^D$. These features are then used inside a Gaussian kernel $\kappa$ on the space $\mathbb{R}^D$, to compute the baseline similarity between data points. We then take a convex combination of that kernel with a Gaussian kernel $q$ on the input space; the weight of this component is determined by a parameter $\epsilon \in (0, 1)$ [7]. The lengthscale of $\kappa$ as well as the mixture parameter $\epsilon$ are included in the overall parameters, $\omega$ or $\gamma$, and learned during the optimization process.

**Overall representation learning algorithm.** The overall procedure, written based on full-batch gradient ascent for simplicity, is shown in Algorithm 1. In practice, we use AdamW (Loshchilov & Hutter, 2019), and draw minibatches in epochs; experimental details are given in Appendix D.1.

**Time complexity.** Let $E_X$ be the cost of computing an embedding $f_\omega(x)$, $E_Y$ be the cost of computing an embedding $f_\gamma(y)$, and $L$ the cost of computing $k_\omega(x, x')$ and $l_\gamma(y, y')$ given the embeddings. Each training iteration costs $\mathcal{O}\left(KE_X + KE_Y + K^2L\right)$, where $K$ is the minibatch size. Typically for practical values of $K$, $E_X + E_Y \gg KL$, so the cost is "almost" linear in practice. [8]

---

[7] Using $\epsilon > 0$ provides a "backup" to the deep kernel, perhaps giving some signal early in optimization when the deep kernel features are not yet useful, and guaranteeing that the overall kernel is characteristic.

[8] Equation (8) could use block estimators (Zaremba et al., 2014) or incomplete $U$-statistics (Blom, 1976) to reduce $\mathcal{O}(K^2L)$ to $\mathcal{O}(K^\beta L)$ for any $\beta \leq 2$, at the cost of increased variance (see Ramdas et al., 2015).

---

**Algorithm 1** HSIC-based independence testing with learned representations

---

**Input:** $S_Z = (S_X, S_Y)$, various hyperparameters used below;

$\omega \leftarrow \omega_0; \gamma \leftarrow \gamma_0; \lambda \leftarrow 10^{-8}$; Split the data into $S_Z^{\text{tr}} \cup S_Z^{\text{te}}; (S_X^{\text{tr}}, S_Y^{\text{tr}}) \leftarrow S_Z^{\text{tr}}; (S_X^{\text{te}}, S_Y^{\text{te}}) \leftarrow S_Z^{\text{te}};$

*# Phase 1: train the kernel parameters $\omega$ and $\gamma$ on $S_Z^{\text{tr}}$*

**for** $T = 1, 2, \ldots, T_{\max}$ **do**

   $Z = (X, Y) \leftarrow$ minibatch from $S_Z^{\text{tr}} = (S_X^{tr}, S_Y^{tr})$;

   $\hat{J}_\lambda(\omega, \gamma) \leftarrow \widehat{\text{HSIC}}_{\text{u}}(X, Y; k_\omega, l_\gamma) \, / \, \hat{\sigma}_{\mathfrak{H}_1}^\lambda(X, Y; k_\omega, l_\gamma);$               *# as in Equation* (8)

   $\omega \leftarrow \omega + \eta \nabla_\omega \hat{J}_\lambda(\omega, \gamma); \quad \gamma \leftarrow \gamma + \eta \nabla_\gamma \hat{J}_\lambda(\omega, \gamma);$            *# maximize $\hat{J}_\lambda(\omega, \gamma)$*

*# Phase 2: permutation test with $k_\omega$ and $l_\gamma$ on $S_Z^{\text{te}}$*

$perm_0 \leftarrow \widehat{\text{HSIC}}_{\text{u}}(S_X^{\text{te}}, S_Y^{\text{te}}; k_\omega, l_\gamma)$                            *# our main estimate*

**for** $i = 1, 2, \ldots, n_{perm}$ **do**

   $perm_i \leftarrow \widehat{\text{HSIC}}_{\text{u}}(\text{shuffle}(S_X^{\text{te}}), S_Y^{\text{te}}; k_\omega, l_\gamma)$               *# no need to shuffle both sets*

**Output:** $k_\omega, l_\gamma, perm_0,$ $p$-value $\frac{1}{n_{perm}} \sum_{i=0}^{n_{perm}} \mathbb{1}(perm_i \geq perm_0)$

---

**MMD-based independence test.** Alternatively to independence tests based on HSIC, we can reformulate the independence problem into a two-sample one, by asking if paired samples $Z = (X, Y)$ and $\tilde{Z} = (\tilde{X}, \tilde{Y})$ drawn from $\mathbb{P}_{xy}$ and $\mathbb{P}_x \times \mathbb{P}_y$ come from the same distribution. To simulate from the null distribution $\mathbb{P}_x \times \mathbb{P}_y$, we can shuffle the samples $Y = (y_1, ..., y_m)$ by some permutation group $\sigma : [m] \to [m]$ so that $\tilde{Y} = (y_{\sigma(1)}, ..., y_{\sigma(m)})$, thereby breaking dependence. With this framing, we can instead maximize the SNR for the corresponding MMD-based two-sample problem, as done by Liu et al. (2020). Although HSIC is theoretically identical to MMD for specific choice of kernels (Proposition 2), we find that the statistical properties of these estimators are different, particularly with respect to how estimation under the null distribution $\mathbb{P}_x \times \mathbb{P}_y$ is performed.

In particular, we notice that MMD estimators based on a single shuffling of the samples exhibit large variance compared to HSIC, which we detail in Appendix C.4. Moreover, as we consider more shufflings of the data, the biased MMD estimator converges to the biased HSIC estimator. This suggests that HSIC estimators handle paired samples more efficiently than its MMD counterpart. Intuitively, this is because HSIC natively considers all possible sample pairings $(x_i, y_j)$ of the data when estimating with respect to $\mathbb{P}_x \times \mathbb{P}_y$, whereas MMD with a single shuffling is only able to consider $m$ such pairs. We explore this phenomenon in more detail in Appendix C.3.

**Theoretical analysis.** Does optimizing the estimator $\hat{J}_\lambda$ of (8) in fact approximately optimize the asymptotic test power $J$ in (7)? Theorem 4 shows that, if the training set size is reasonably large and we can optimize the estimates successfully, we can learn a kernel that generalizes nearly optimally and is hence powerful for large $m$, not just overfitting to the training set (and finding a bad test).

**Theorem 4.** *Under Assumptions (A) to (C), let $\Theta \subseteq \Omega \times \Gamma$ be a set of kernel parameters $\theta \in \Theta$ for which $\sigma_\theta^2 \geq s^2$, and $n$ be the training set size; take $\lambda = \Theta\left(n^{-1/3}\right)$. Then*

$$\sup_{\theta := (\omega, \gamma) \in \Theta} \left| \hat{J}_\lambda(X, Y; k_\omega, l_\gamma) - J(X, Y; k_\omega, l_\gamma) \right| = \tilde{\mathcal{O}}_p\left( \frac{1}{s^2 n^{1/3}} \left[ \frac{1}{s} + L_k + L_l + \sqrt{D_\Omega} + \sqrt{D_\Gamma} \right] \right).$$

*If there are unique best kernels $(k_{\omega^*}, l_{\gamma^*})$ maximizing $J(X, Y; k_\omega, l_\gamma)$, then the maximizer of $\hat{J}_\lambda(X, Y; k_\omega, l_\gamma)$ converges in probability to $(k_{\omega^*}, l_{\gamma^*})$ as $n \to \infty$.*

Appendix B states and proves a nonasymptotic version of this result; the assumptions and proof techniques are based on those of Liu et al. (2020).

## 6 EXPERIMENTS

**Baselines**. We compare our HSIC-based (HSIC-D/Dx/O) and MI-based (InfoNCE/NWJ) tests with various baselines. All tests are performed via permutation testing.

- HSIC-D: HSIC using deep kernels on each space $\mathcal{X}$ and $\mathcal{Y}$; simultaneously trained via Section 5.
- HSIC-Dx: HSIC using a tied deep kernel, i.e. $k_\omega = l_\gamma$, and trained via Section 5.
- HSIC-O: HSIC using Gaussian kernels, with each bandwidth parameter optimized via Section 5.

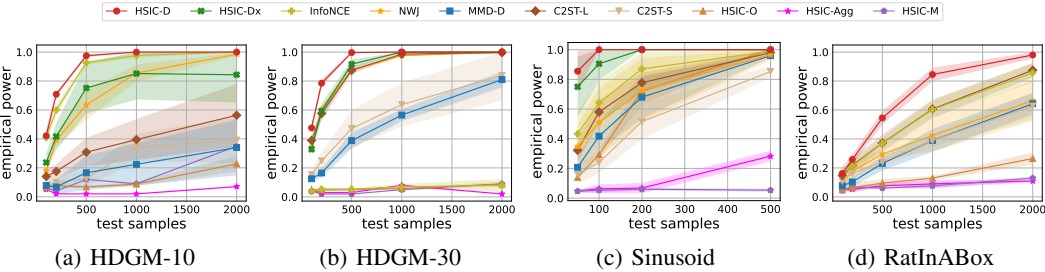

(a) HDGM-10 (b) HDGM-30 (c) Sinusoid (d) RatInABox

Figure 2: Empirical power vs sample size $m$ for different datasets, when trained with a large training set. The average test power is computed over 5 training runs, where the empirical power is determined over 100 permutation tests. The shaded region covers one standard error from the mean.

- HSIC-M: HSIC using Gaussian kernels, with bandwidth selected via the median heuristic.
- HSIC-Agg (Schrab et al., 2023): aggregating Gaussian kernels, with their default settings.
- MMD-D: The method of Liu et al. (2020) applied to $\mathbb{P}_{xy}$ vs $\mathbb{P}_x \times \mathbb{P}_y$,[9] with a kernel on $\mathcal{X} \times \mathcal{Y}$.
- C2ST-S (Lopez-Paz & Oquab, 2017) / C2ST-L (Cheng & Cloninger, 2022): Sign/logit-based classifier two-sample test for $\mathbb{P}_{xy}$ vs $\mathbb{P}_x \times \mathbb{P}_y$.
- InfoNCE (van den Oord et al., 2018; Poole et al., 2019): the statistic $\hat{I}_{\mathrm{NCE}}$ as in (1).
- NWJ (Nguyen et al., 2010; Poole et al., 2019): another mutual information bound statistic $\hat{I}_{\mathrm{NWJ}}$.

**Datasets.** We consider three informative synthetic datasets, where the true answers are known.

• **High-dimensional Gaussian mixture**. The distribution HDGM-$d$ has $d$ total dimensions (divided between $X$ and $Y$), but has dependence only between two of them:

$$\left[X_1, Y_{\lfloor d/2 \rfloor}, \ldots, X_{\lceil d/2 \rceil}, Y_1\right] \sim \sum_{i=1}^{2} \frac{1}{2} \mathcal{N}\left(\mathbf{0}_d, \begin{bmatrix} 1 & 0.5(-1)^i & \mathbf{0}_{d-2}^T \\ 0.5(-1)^i & 1 & \mathbf{0}_{d-2}^T \\ \mathbf{0}_{d-2} & \mathbf{0}_{d-2} & I_{d-2} \end{bmatrix}\right),$$

where the odd dimensions are taken to be from $\mathbb{P}_x$ and even dimensions to be from $\mathbb{P}_y$. Moreover, for $d \geq 4$ the dependent variables $X_1$ and $Y_{\lfloor d/2 \rfloor}$ are at different dimensions. We perform independence tests at dimensions 4, 8, 10, 20, 30, 40, and 50.

• **Sinusoid** (Sejdinovic et al., 2012). We sample from sinusoidally dependent data with distribution $\mathbb{P}_{xy} \propto 1 + \sin(\ell x)\sin(\ell y)$ on support $\mathcal{X} \times \mathcal{Y} = [-\pi, \pi]^2$. Higher frequencies $\ell$ produce subtler departures from the uniform distribution, resulting in a harder independence problem; we use $\ell = 4$. A visualization of this density is given in Figure 10.

• **RatInABox** (George et al., 2024). RatInABox simulates hippocampal cells of a rat in motion. In particular, we test for dependence between firing rates of grid cells and the rat's head direction. Grid cells respond near points in a grid covering the environment surface, and should be subtly connected to head direction because of the geometry of the "box" (Figure 9). We consider 8 grid cells, and simulate motion for 100 000 seconds, taking a measurement every 5 seconds as our dataset.

**Power versus test size**. We first compare how well methods identify dependency with a large available training set, by comparing the rate at which the learned tests achieve perfect power (1.0) as the test set size $m$ increases. Results on the HDGM problems are given in Figures 1 and 5. We use a training size of 10,000 for HDGM $\leq 30$, 100,000 for HDGM > 30, and 2000 validation samples for all dimensions. For the Sinusoid problem, we train all optimization-based tests on 5,000 samples and use 1,000 for validation; the results are shown in Figure 2 (c). RatInABox results are shown in Figure 2 (d); we use a training size of 4,000 samples with no validation.

Overall, HSIC-D outperforms baselines, and is able to reach perfect power at smaller test sizes $m$. We note that HSIC-O perform reasonably well for simpler problems like Sinusoid, but struggle to on harder problems like RatInABox. This suggests that no Gaussian kernel on the input data is well-suited for the task. On the other hand, such kernels applied to optimal representations of the data (HSIC-D) are the most powerful.

---

[9]Specifically, we compare the given samples to a single shuffling of the given samples.

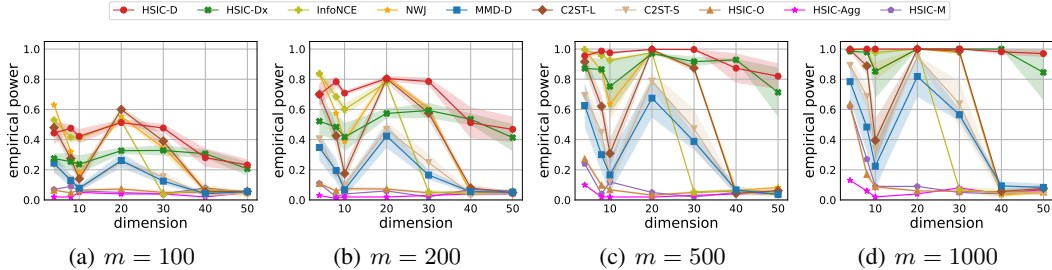

Figure 3: Empirical power vs dimension across various test sample sizes $m = \{100, 200, 500, 1000\}$ for HDGM. The shaded region covers one standard error over 5 training runs.

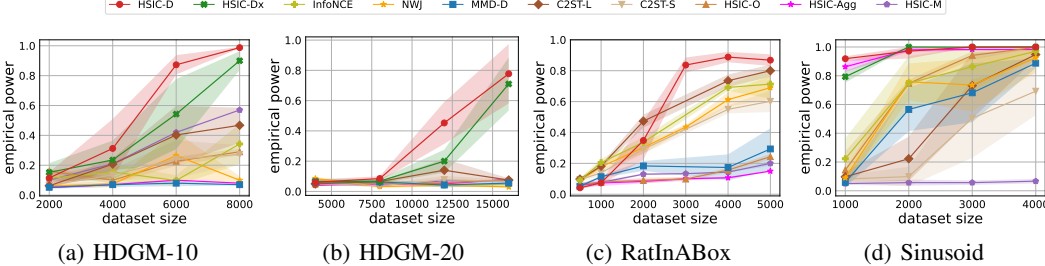

Figure 4: Empirical power vs dataset size. HDGM and Sinusoid uses a consistent 7:2:1 train-val-test split across all dataset sizes, while RatInABox maintains a 3:2 train-test split. HISC-Agg and HSIC-M do not split the data. The shaded region covers one standard deviation over 5 training runs.

We also note that MMD-D underperforms HSIC-D in almost all settings, despite their equivalence under certain kernel functions. We attribute this to the increased variance of the MMD estimator when using only a single shuffling of the data, detailed in Appendix C.3.

**Power versus dimension**. We demonstrate the effectiveness of our method at various dimensions $d/2$ by examining the empirical test power at HDGM-$d$ for $d \in \{4, 8, 10, 20, 30, 40, 50\}$ with fixed test sizes $m$. We use the same training splits as before. Results are shown in Figure 3. Overall, HSIC-D exhibits the highest test power across all dimensions. When a small number of test samples are used (i.e., $m = 100$) the performance of HSIC-D slightly degrades with increasing dimension, while at larger test sample sizes it consistently has near-perfect power.

**Power versus dataset size**. A drawback to kernel selection via optimization is that we must hold out a split of the data for training. In contrast, HSIC-M and HSIC-Agg are able to utilize the entirety of the dataset for their test. To examine this trade-off, we consider consistent data splitting at datasets of varying sizes. For the HDGM and Sinusoid problems we use a 7:2:1 train-val-test split, and for RatInABox we use a 3:2 train-test split. Conversely, HSIC-M and HSIC-Agg use the *entire* dataset for testing. Results are shown in Figure 4. Heuristic methods are able to take advantage of the additional test samples and outperform some data-splitting methods at smaller dataset sizes; for large dataset sizes the reverse is true.

**Optimizing J versus optimizing HSIC**. Rather than optimizing objective $J$, we can maximize the test statistic HSIC; however, doing so makes no guarantees on the test power. Figure 8 compares these objectives and demonstrates that optimizing $J$ is preferred over optimizing the HSIC.

## 7 CONCLUSION AND FUTURE WORK

Independence testing aims to see if two paired random variables are statistically independent. We explored two families of tests to address this problem. The first are tests based on variational mutual information estimators, which, to the best of our knowledge, we are the first to construct. The second are tests based on maximizing the asymptotic power, which was explored for the two-sample problem but not for independence testing. Our findings show that learning representations of the data via our proposed methods lead to powerful tests, with HSIC-based tests generally outperforming MI-based ones. Future work may look to extend this learning scheme to conditional independence testing and apply this to causal discovery. Meanwhile, we may also look into approaches for mitigating, or even removing, the data-splitting procedure as done by Biggs et al. (2023) and Kübler et al. (2020).

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

## A ASYMPTOTICS OF HSIC ESTIMATORS

**Proposition 5** (Song et al., 2012, Theorem 5). *Under the alternative hypothesis $\mathfrak{H}_1 : \mathbb{P}_{xy} \neq \mathbb{P}_x \times \mathbb{P}_y$, the unbiased estimator of HSIC is asymptotically normal: with $m$ samples,*

$$\sqrt{m}(\widehat{\text{HSIC}}_u - \text{HSIC}) \xrightarrow{d} \mathcal{N}(0, \sigma^2_{\mathfrak{H}_1}). \tag{9}$$

*The asymptotic variance of $\sqrt{m}\,\widehat{\text{HSIC}}_u$, $\sigma^2_{\mathfrak{H}_1}$, can be consistently estimated from $n$ samples[10] as $16\left(R - \text{HSIC}^2\right)$, where $R = \frac{1}{n}\sum_{i=1}^n \left(\frac{(n-4)!}{(n-1)!}\sum_{(j,q,r)\in\mathbf{i}_3^n / \{i\}} h(i,j,q,r)\right)^2$. Here $\mathbf{i}_n^\ell \setminus \{i\}$ denotes the set of all $\ell$-tuples drawn without replacement from the set $\{1,\ldots,n\}\setminus\{i\}$, and $h(i,j,q,r) = \frac{1}{24}\sum_{(s,t,u,v)}^{(i,j,q,r)} k_{st}(l_{st}+l_{uv}-2l_{su})$, where the sum ranges over all $4! = 24$ ways to assign the distinct indices $\{i,j,q,r\}$ to the four variables $(s,t,u,v)$ without replacement.*

The behavior of the estimator is different under the null hypothesis ($X \perp\!\!\!\perp Y$); in this regime $m\,\widehat{\text{HSIC}}_u$ (rather than $\sqrt{m}\widehat{\text{HSIC}}_u$) converges in distribution to something with complex dependence on $\mathbb{P}_x$, $\mathbb{P}_y$, $k$, and $l$.

**Proposition 6** (Gretton et al., 2007, Theorem 2). *Under the null hypothesis $\mathfrak{H}_0 : \mathbb{P}_{xy} = \mathbb{P}_x \times \mathbb{P}_y$, the U-statistic estimator of HSIC is degenerate. In this case $m\widehat{\text{HSIC}}_u$ converges in distribution to*

$$m\widehat{\text{HSIC}}_u \xrightarrow{d} \sum_{\ell=1}^\infty \lambda_\ell(z_\ell^2 - 1),$$

*where $z_\ell \overset{iid}{\sim} \mathcal{N}(0,1)$, and $\lambda_\ell$ are the solutions to the eigenvalue problem*

$$\lambda_\ell \psi_\ell(z_j) = \int h_{ijqr}\psi_\ell(z_i)\,\mathrm{d}F_{i,q,r},$$

*where $h_{ijqr} := h(i,j,q,r)$ is defined in Proposition 5, and $F_{i,q,r}$ denotes the probability measure with respect to variables $z_i$, $z_q$, and $z_r$.*

## B UNIFORM CONVERGENCE ANALYSIS

### B.1 PRELIMINARIES

We start by defining the notation used in the proofs. Let kernels $k_\omega$ and $l_\gamma$ be parameterized by some $\omega$ and $\gamma$, and samples $(X_i, Y_i) \sim \mathbb{P}_{xy}$ drawn i.i.d. from the joint distribution. We denote the $n \times n$ gram matrices of $k_\omega$ and $l_\gamma$ by $K^{(\omega)}$ and $L^{(\gamma)}$ respectively. We will often omit the kernel parameters $\omega$ and $\gamma$ when it is clear from the context.

Let $\eta$ be the HSIC test statistic and $\hat\eta$ to be its U-statistic estimator given by

$$\hat\eta = \frac{1}{(n)_4}\sum_{(i,j,q,r)\in\mathbf{i}_4^n} H_{ijqr},$$

where $(n)_k = n!/(n-k)!$ is the Pochhammer symbol and $\mathbf{i}_4^n$ is all possible 4-tuples drawn without replacement from 1 to n. $H$ is the kernel gram matrix of the U-statistic defined by

$$H_{ijqr} = \frac{1}{4!}\sum_{(a,b,c,d)}^{(i,j,q,r)} K_{ab}\left(L_{ab} + L_{cd} - 2L_{ac}\right),$$

where sum represents all 4! combinations of tuples $(a,b,c,d)$ that can be selected without replacement from $(i,j,q,r)$.

---

[10]Typically $m = n$, but we might want to use a few ($n$) samples to roughly estimate the power of an $m$-sample test with $m \gg n$, as done in a different context by Sutherland & Deka (2019); Deka & Sutherland (2023).

## B.2 MAIN RESULTS

Our main uniform convergence results require the following assumptions.

(A) The set of kernel parameters $\Omega$ lies in a Banach space of dimension $D_\Omega$, and the set of kernel parameters $\Gamma$ lies in a Banach space of dimension $D_\Gamma$. Furthermore, each parameter space is bounded by $R_\Omega$ and $R_\Gamma$ respectively, i.e.,

$$\Omega \subseteq \{\omega \mid \|\omega\| \leq R_\Omega\},$$
$$\Gamma \subseteq \{\gamma \mid \|\gamma\| \leq R_\Gamma\}.$$

(B) The kernels $k_\omega$ and $l_v$ are uniformly bounded:

$$\sup_{\omega \in \Omega} \sup_{x \in \mathcal{X}} k_\omega(x, x) \leq \nu_k,$$
$$\sup_{\gamma \in \Gamma} \sup_{x \in \mathcal{Y}} l_\gamma(y, y) \leq \nu_l.$$

For the kernels we use in practice, $\nu_k = \nu_l = 1$.

(C) Both kernels $k$ and $l$ are Lipschitz with respect to the parameter space: for all $x, x' \in \mathcal{X}$ and $\omega, \omega' \in \Omega$

$$|k_\omega(x, x') - k_{\omega'}(x, x')| \leq L_k \|\omega - \omega'\|,$$

and for all $y, y' \in \mathcal{Y}$ and $\gamma, \gamma' \in \Gamma$

$$|l_\gamma(y, y') - l_{\gamma'}(y, y')| \leq L_l \|\gamma - \gamma'\|.$$

**Theorem 7.** *Under Assumptions (A) to (C), let $\Theta \subseteq \Omega \times \Gamma$ be the set of kernel parameters $\theta \in \Theta$ for which $\sigma_\theta^2 \geq s^2$, and take $\lambda = n^{-1/3}$. Assume $\nu_k, \nu_l \geq 1$. Then, with probability at least $1 - \delta$,*

$$\sup_{\theta \in \Theta} \left| \frac{\hat{\eta}_\theta}{\hat{\sigma}_{\theta,\lambda}} - \frac{\eta_\theta}{\sigma_\theta} \right| \leq \frac{2\nu_k\nu_l}{s^2 n^{1/3}} \left[ \frac{1}{s} + \frac{9216\nu_k^2\nu_l^2}{\sqrt{n}} \right.$$

$$\left. + \left( 12288\nu_k^2\nu_l^2 + \frac{8s}{n^{1/6}} \right) \left( \frac{L_k}{\nu_k} + \frac{L_l}{\nu_l} + \sqrt{2\log\frac{4}{\delta} + 2D_\Omega \log(4R_\Omega\sqrt{n}) + 2D_\Gamma \log(4R_\Gamma\sqrt{n})} \right) \right],$$

*and thus, treating $\nu_k, \nu_l$ as constants,*

$$\sup_{\theta \in \Theta} \left| \frac{\hat{\eta}_\theta}{\hat{\sigma}_{\theta,\lambda}} - \frac{\eta_\theta}{\sigma_\theta} \right| = \tilde{\mathcal{O}}_P \left( \frac{1}{s^2 n^{1/3}} \left[ \frac{1}{s} + L_k + L_l + \sqrt{D_\Omega} + \sqrt{D_\Gamma} \right] \right).$$

*Proof.* Let $\hat{\sigma}_{\theta,\lambda}^2 := \hat{\sigma}_\theta^2 + \lambda$ be our regularized variance estimator from which we can assume is positive. We start by decomposing

$$\sup_{\theta \in \Theta} \left| \frac{\hat{\eta}_\theta}{\hat{\sigma}_{\theta,\lambda}} - \frac{\eta_\theta}{\sigma_\theta} \right| \leq \sup_{\theta \in \Theta} \left| \frac{\hat{\eta}_\theta}{\hat{\sigma}_{\theta,\lambda}} - \frac{\hat{\eta}_\theta}{\sigma_{\theta,\lambda}} \right| + \sup_{\theta \in \Theta} \left| \frac{\hat{\eta}_\theta}{\sigma_{\theta,\lambda}} - \frac{\hat{\eta}_\theta}{\sigma_\theta} \right| + \sup_{\theta \in \Theta} \left| \frac{\hat{\eta}_\theta}{\sigma_\theta} - \frac{\eta_\theta}{\sigma_\theta} \right|$$

$$= \sup_\theta \frac{|\hat{\eta}_\theta|}{\hat{\sigma}_{\theta,\lambda} \cdot \sigma_{\theta,\lambda}} \frac{|\hat{\sigma}_{\theta,\lambda}^2 - \sigma_{\theta,\lambda}^2|}{\hat{\sigma}_{\theta,\lambda} + \sigma_{\theta,\lambda}} + \sup_\theta \frac{|\hat{\eta}_\theta|}{\sigma_{\theta,\lambda} \cdot \sigma_\theta} \frac{|\sigma_{\theta,\lambda}^2 - \sigma_\theta^2|}{\sigma_{\theta,\lambda} + \sigma_\theta} + \sup_\theta \frac{1}{\sigma_\theta} |\hat{\eta}_\theta - \eta_\theta|$$

$$\leq \frac{4\nu_k\nu_l}{s\sqrt{\lambda}(s + \sqrt{\lambda})} \sup_\theta |\hat{\sigma}_\theta^2 - \sigma_\theta^2| + \frac{4\nu_k\nu_l\lambda}{s\sqrt{s^2 + \lambda}(s + \sqrt{s^2 + \lambda})} + \frac{1}{s} \sup_\theta |\hat{\eta}_\theta - \eta_\theta|$$

$$\leq \frac{4\nu_k\nu_l}{s^2\sqrt{\lambda}} \sup_\theta |\hat{\sigma}_\theta^2 - \sigma_\theta^2| + \frac{1}{s} \sup_\theta |\hat{\eta}_\theta - \eta_\theta| + \frac{2\nu_k\nu_l\lambda}{s^3}.$$

Proposition 8 and Proposition 9 show the uniform convergence of $\hat{\eta}_\theta$ and $\hat{\sigma}_\theta$, from which we get that with probability at least $1 - \delta$, the error is at most

$$\sup_{\theta \in \Theta} \left| \frac{\hat{\eta}_\theta}{\hat{\sigma}_{\theta,\lambda}} - \frac{\eta_\theta}{\sigma_\theta} \right| \leq \frac{2\nu_k\nu_l\lambda}{s^3} + \frac{18432\nu_k^3\nu_l^3}{s^2 n\sqrt{\lambda}} + \left[ \frac{8192\nu_k^3\nu_l^3}{s^2\sqrt{\lambda}n} + \frac{8\nu_k\nu_l}{s\sqrt{n}} \right] \left( \frac{L_k}{\nu_k} + \frac{L_l}{\nu_l} \right)$$

$$+ \left[ \frac{24576\nu_k^3\nu_l^3}{s^2\sqrt{\lambda}n} + \frac{16\nu_k\nu_l}{s\sqrt{n}} \right] \sqrt{2\log\frac{4}{\delta} + 2D_\Omega \log(4R_\Omega\sqrt{n}) + 2D_\Gamma \log(4R_\Gamma\sqrt{n})}.$$

Taking $\lambda = n^{-1/3}$ gives

$$\sup_{\theta \in \Theta} \left| \frac{\hat{\eta}_\theta}{\hat{\sigma}_{\theta,\lambda}} - \frac{\eta_\theta}{\sigma_\theta} \right| \leq \frac{2\nu_k \nu_l}{s^3 n^{1/3}} + \frac{18432 \nu_k^3 \nu_l^3}{s^2 n^{5/6}} + \left[ \frac{8192 \nu_k^3 \nu_l^3}{s^2 n^{1/3}} + \frac{8\nu_k \nu_l}{s\sqrt{n}} \right] \left( \frac{L_k}{\nu_k} + \frac{L_l}{\nu_l} \right)$$
$$+ \left[ \frac{24576 \nu_k^3 \nu_l^3}{s^2 n^{1/3}} + \frac{16\nu_k \nu_l}{s\sqrt{n}} \right] \sqrt{2 \log \frac{4}{\delta} + 2D_\Omega \log(4R_\Omega \sqrt{n}) + 2D_\Gamma \log(4R_\Gamma \sqrt{n})}.$$

Using $\nu_k, \nu_l \geq 1$ we can slightly simplify our bound to

$$\sup_{\theta \in \Theta} \left| \frac{\hat{\eta}_\theta}{\hat{\sigma}_{\theta,\lambda}} - \frac{\eta_\theta}{\sigma_\theta} \right| \leq \left[ \frac{24576 \nu_k^3 \nu_l^3}{s^2 n^{1/3}} + \frac{16\nu_k \nu_l}{s\sqrt{n}} \right] \left( \frac{L_k}{\nu_k} + \frac{L_l}{\nu_l} + \sqrt{2 \log \frac{4}{\delta} + 2D_\Omega \log(4R_\Omega \sqrt{n}) + 2D_\Gamma \log(4R_\Gamma \sqrt{n})} \right)$$
$$+ \frac{2\nu_k \nu_l}{s^3 n^{1/3}} + \frac{18432 \nu_k^3 \nu_l^3}{s^2 n^{5/6}}$$

$\square$

## B.3 Uniform convergence results

This subsection pertains to uniform convergence results of $\hat{\eta}_\theta$ and $\hat{\sigma}_\theta$, and are are used in the proof of Theorem 7.

**Proposition 8.** *Under assumptions (A) to (C), we have that with probability at least $1 - \delta$,*

$$\sup_{\theta \in \Theta} |\hat{\eta}_\theta - \eta_\theta| \leq \frac{8\nu_k \nu_l}{\sqrt{n}} \left( \frac{L_k}{\nu_k} + \frac{L_l}{\nu_l} + 2\sqrt{2 \log \frac{2}{\delta} + 2D_\Omega \log(4R_\Omega \sqrt{n}) + 2D_\Gamma \log(4R_\Gamma \sqrt{n})} \right).$$

*Proof.* We use $\epsilon$-net arguments on both spaces $\Omega$ and $\Gamma$. Let $\{\omega_i\}_{i=1}^{T_\Omega}$ be arbitrarily placed centers with radius $\rho_\Omega$ such that any point $\omega \in \Omega$ satisfies $\min \|\omega - \omega_i\| \leq \rho_\Omega$. Similarly, let $\{\gamma_i\}_{i=1}^{T_\Gamma}$ be centers with radius $\rho_\Gamma$ satisfying $\min \|\gamma - \gamma_i\| \leq \rho_\Gamma$ for any $\gamma \in \Gamma$. Assumption (B) ensures this is possible with at most $T_\Omega = (4R_\Omega/\rho_\Omega)^{D_\Omega}$ and $T_\Gamma = (4R_\Gamma/\rho_\Gamma)^{D_\Gamma}$ points respectively (Cucker & Smale, 2002, Proposition 5).

We can decompose the convergence bound into simpler components and tackle each component individually

$$\sup_{\theta \in \Theta} |\hat{\eta}_\theta - \eta_\theta| \leq \sup_\theta |\hat{\eta}_\theta - \hat{\eta}_{\theta'}| + \max_{\substack{\omega' \in \{\omega_1, \ldots, \omega_{T_\Omega}\} \\ \gamma' \in \{\gamma_1, \ldots, \gamma_{T_\Gamma}\}}} |\hat{\eta}_{\theta'} - \eta_{\theta'}| + \sup_\theta |\eta_{\theta'} - \eta_\theta|.$$

First, let us analyze $|\eta_\theta - \eta_{\theta'}|$ for any $\theta, \theta' \in \Theta$. Recall that $\eta = \mathbb{E}[H_{1234}]$ where $H_{1234} = \frac{1}{4!} \sum_{(a,b,c,d)}^{(1,2,3,4)} K_{ab}(L_{ab} + L_{cd} - 2L_{ac})$. We have that

$$|H_{1234}^{(\theta)} - H_{1234}^{(\theta')}| \leq \frac{1}{4!} \sum_{(abcd)}^{(1234)} \left| K_{ab}^{(\omega)}(L_{ab}^{(\gamma)} + L_{cd}^{(\gamma)} - 2L_{ac}^{(\gamma)}) - K_{ab}^{(\omega')}(L_{ab}^{(\gamma')} + L_{cd}^{(\gamma')} - 2L_{ac}^{(\gamma')}) \right|$$

$$\leq \frac{1}{4!} \sum_{(abcd)}^{(1234)} \left( \left| K_{ab}^{(\omega)} L_{ab}^{(\gamma)} - K_{ab}^{(\omega')} L_{ab}^{(\gamma')} \right| + \left| K_{ab}^{(\omega)} L_{cd}^{(\gamma)} - K_{ab}^{(\omega')} L_{cd}^{(\gamma')} \right| + 2 \left| K_{ab}^{(\omega')} L_{ac}^{(\gamma')} - K_{ab}^{(\omega)} L_{ac}^{(\gamma)} \right| \right).$$

From Assumption (A) we know that $|K_{ab}| \leq \nu_k$ and $|L_{ab}| \leq \nu_l$, and via Assumption (C) we notice that

$$\left| K_{ab}^{(\omega)} L_{ab}^{(\gamma)} - K_{ab}^{(\omega')} L_{ab}^{(\gamma')} \right| = \left| K_{ab}^{(\omega)} L_{ab}^{(\gamma)} - K_{ab}^{(\omega)} L_{ab}^{(\gamma')} + K_{ab}^{(\omega)} L_{ab}^{(\gamma')} - K_{ab}^{(\omega')} L_{ab}^{(\gamma')} \right|$$

$$\leq \left| K_{ab}^{(\omega)} \right| \left| L_{ab}^{(\gamma)} - L_{ab}^{(\gamma')} \right| + \left| L_{ab}^{(\gamma')} \right| \left| K_{ab}^{(\omega)} - K_{ab}^{(\omega')} \right|$$

$$\leq \nu_k L_l \|v - v'\| + \nu_l L_k \|\omega - \omega'\|$$

$$\leq \nu_k L_l \rho_\Gamma + \nu_l L_k \rho_\Omega.$$

This expression is true for all three components of $|H_{1234}^{(\theta)} - H_{1234}^{(\theta')}|$ and so it follows that

$$|\eta_\theta - \eta_{\theta'}| = \left|\mathbb{E}[H_{1234}^{(\theta)}] - \mathbb{E}[H_{1234}^{(\theta')}]\right| \leq \mathbb{E}\left|H_{1234}^{(\theta)} - H_{1234}^{(\theta')}\right| \leq 4\nu_k L_l \rho_\Gamma + 4\nu_l L_k \rho_\Omega,$$

$$|\hat\eta_\theta - \hat\eta_{\theta'}| = \left|\frac{1}{(n)_4} \sum_{(i,j,q,r) \in \mathbf{i}_4^n} H_{ijqr}^{(\theta)} - H_{ijqr}^{(\theta')}\right| \leq \frac{1}{(n)_4} \sum_{(i,j,q,r) \in \mathbf{i}_4^n} \left|H_{1234}^{(\theta)} - H_{1234}^{(\theta')}\right| \leq 4\nu_k L_l \rho_\Gamma + 4\nu_l L_k \rho_\Omega.$$

Now, we study the random error function $\Delta := \hat\eta - \eta$. Note that $\mathbb{E}\,\Delta = 0$ since $\hat\eta$ is unbiased, and $|H_{ijqr}| \leq 4\nu_k\nu_l$ via Assumption (A). This $\hat\eta$, and hence $\Delta$, satisfies bounded differences. Let $F$ denote the kernel tensor $H$ but with sample $(X_\ell, Y_\ell)$ replaced by $(X'_\ell, Y'_\ell)$ so that $F$ agrees with $H$ except at indicies $\ell$, and let $\hat\eta' = \frac{1}{(n)_4} \sum_{(i,j,q,r) \in \mathbf{i}_4^n} F_{ijqr}$ be it's HSIC estimator.

For convenience, we denote $(i,j,q,r) \in \mathbf{i}_4^n$ simply as $(i,j,q,r)$, and $(i,j,q)\backslash k$ to be the set of 3-tuples drawn without replacement from $\mathbf{i}_3^n$ that exclude the number $k$. We can compute the maximal bounded difference $|\Delta - \Delta'| = |\hat\eta - \hat\eta'|$ as

$$|\hat\eta - \hat\eta'| = \left|\frac{1}{(n)_4} \sum_{(i,j,q,r)} H_{ijqr} - F_{ijqr}\right| \leq \frac{1}{(n)_4} \sum_{(i,j,q,r)} |H_{ijqr} - F_{ijqr}| \tag{10}$$

$$= \frac{1}{(n)_4}\left(\sum_{(j,q,r)\backslash\ell} \underbrace{|H_{\ell jqr} - F_{\ell jqr}|}_{\leq 8\nu_k\nu_l} + \sum_{(i,q,r)\backslash\ell} |H_{i\ell qr} - F_{i\ell qr}| + \sum_{(i,j,r)\backslash\ell} |H_{ij\ell r} - F_{ij\ell r}| + \sum_{(i,j,q)\backslash\ell} |H_{ijq\ell} - F_{ijq\ell}|\right)$$

$$= \frac{1}{(n)_4}\left((n-1)_3 \cdot 8\nu_k\nu_l \cdot 4\right) = \frac{32\nu_k\nu_l}{n}.$$

Then, applying McDiarmid's inequality on $\Delta := \hat\eta - \eta$ followed by a union bound over the $T_\Omega T_\Gamma$ center pairs gives us, with probability at least $1 - \delta$, that

$$\max_{\substack{\omega' \in \{\omega_1, \ldots, \omega_{T_\Omega}\} \\ \gamma' \in \{\gamma_1, \ldots, \gamma_{T_\Gamma}\}}} |\hat\eta_{\theta'} - \eta_{\theta'}| \leq 32\nu_k\nu_l \sqrt{\frac{1}{2n} \log \frac{2T_\Omega T_\Gamma}{\delta}}$$

$$= \frac{16\nu_k\nu_l}{\sqrt{n}} \sqrt{2\log\frac{2}{\delta} + 2\log T_\Omega + 2\log T_\Gamma}$$

$$= \frac{16\nu_k\nu_l}{\sqrt{n}} \sqrt{2\log\frac{2}{\delta} + 2D_\Omega \log\frac{4R_\Omega}{\rho_\Omega} + 2D_\Gamma \log\frac{4R_\Gamma}{\rho_\Gamma}}.$$

Finally, we combine these results to get our uniform convergence bound:

$$\sup_{\theta \in \Theta} |\hat\eta_\theta - \eta_\theta| \leq 8\nu_k L_l \rho_\Gamma + 8\nu_l L_k \rho_\Omega + \frac{16\nu_k\nu_l}{\sqrt{n}} \sqrt{2\log\frac{2}{\delta} + 2D_\Omega \log\frac{4R_\Omega}{\rho_\Omega} + 2D_\Gamma \log\frac{4R_\Gamma}{\rho_\Gamma}}$$

$$= 8\nu_k\nu_l \left(\frac{L_k}{\nu_k}\rho_\Omega + \frac{L_l}{\nu_l}\rho_\Gamma + \frac{2}{\sqrt{n}}\sqrt{2\log\frac{2}{\delta} + 2D_\Omega \log\frac{4R_\Omega}{\rho_\Omega} + 2D_\Gamma \log\frac{4R_\Gamma}{\rho_\Gamma}}\right).$$

Setting $\rho_\Omega = \rho_\Gamma = 1/\sqrt{n}$ yields the desired result. $\qquad\square$

**Proposition 9.** *Under assumptions (A) to (C), we have that with probability at least $1 - \delta$,*

$$\sup_{\theta \in \Theta} |\hat\sigma_\theta^2 - \sigma_\theta^2| \leq \frac{2048\nu_k^2\nu_l^2}{\sqrt{n}}\left(\frac{L_k}{\nu_k} + \frac{L_l}{\nu_l} + 3\sqrt{2\log\frac{2}{\delta} + 2D_\Omega \log(4R_\Omega\sqrt{n}) + 2D_\Gamma \log(4R_\Gamma\sqrt{n})} + \frac{9}{4\sqrt{n}}\right).$$

*Proof.* We use an $\epsilon$-net argument on both spaces $\Omega$ and $\Gamma$. Using the same construction as in Proposition 8, we once again decompose our convergence bound:

$$\sup_{\theta \in \Theta} |\hat\sigma_\theta^2 - \sigma_\theta^2| \leq \sup_\theta |\hat\sigma_\theta^2 - \hat\sigma_{\theta'}^2| + \max_{\substack{\omega' \in \{\omega_1, \ldots, \omega_{T_\Omega}\} \\ \gamma' \in \{\gamma_1, \ldots, \gamma_{T_\Gamma}\}}} |\hat\sigma_{\theta'}^2 - \sigma_{\theta'}^2| + \sup_\theta |\sigma_{\theta'}^2 - \sigma_\theta^2|.$$

First, let us analyze $|\sigma_\theta^2 - \sigma_{\theta'}^2|$ for any $\theta, \theta' \in \Theta$. Recall that $\sigma^2 = 16\left(\mathbb{E}[H_{1234}H_{1567}] - \eta^2\right)$. It follows that

$$|\sigma_\theta^2 - \sigma_{\theta'}^2| = 16\left|\mathbb{E}[H_{1234}^{(\theta)}H_{1567}^{(\theta)} - H_{1234}^{(\theta')}H_{1567}^{(\theta')}] - \mathbb{E}[H_{1234}^{(\theta)}H_{5678}^{(\theta)}] + \mathbb{E}[H_{1234}^{(\theta')}H_{5678}^{(\theta')}]]\right|$$

$$\leq 16\,\mathbb{E}\left|H_{1234}^{(\theta)}H_{1567}^{(\theta)} - H_{1234}^{(\theta')}H_{1567}^{(\theta')}\right| + 16\,\mathbb{E}\left|H_{1234}^{(\theta)}H_{5678}^{(\theta)} - H_{1234}^{(\theta')}H_{5678}^{(\theta')}\right|.$$

Under Assumptions (A) and (C) we know that $|H_{1234}| \leq 4\nu_k\nu_l$ and $|H_{1234}^{(\theta)} - H_{1234}^{(\theta')}| \leq 4\nu_k L_l\rho_\Gamma + 4\nu_l L_k\rho_\Omega$. As such, we have

$$|H_{1234}^{(\theta)}H_{1567}^{(\theta)} - H_{1234}^{(\theta')}H_{1567}^{(\theta')}| \leq |H_{1234}^{(\theta)}H_{1567}^{(\theta)} - H_{1234}^{(\theta)}H_{1567}^{(\theta')}| + |H_{1234}^{(\theta)}H_{1567}^{(\theta')} - H_{1234}^{(\theta')}H_{1567}^{(\theta')}|$$

$$= |H_{1234}^{(\theta)}||H_{1567}^{(\theta)} - H_{1567}^{(\theta')}| + |H_{1567}^{(\theta')}||H_{1234}^{(\theta)} - H_{1234}^{(\theta')}|$$

$$\leq 32\nu_k\nu_l(\nu_k L_l\rho_\Gamma + \nu_l L_k\rho_\Omega)$$

This expression is true for both components of $|\sigma_\theta^2 - \sigma_{\theta'}^2|$ and so it follows that

$$|\sigma_\theta^2 - \sigma_{\theta'}^2| \leq 1024\nu_k\nu_l(\nu_k L_l\rho_\Gamma + \nu_l L_k\rho_\Omega). \tag{11}$$

Similarly, replacing the expectations $\mathbb{E}[H_{1234}H_{1567}]$ and $\mathbb{E}[H_{1234}H_{5678}]$ with the respective estimators $\frac{1}{(n)_4(n-1)_3}\sum_{(ijqr),(bcd)\setminus i} H_{ijqr}H_{ibcd}$ and $\frac{1}{(n)_4^2}\sum_{(ijqr),(abcd)} H_{ijqr}H_{abcd}$ give us the same bound

$$|\hat\sigma_\theta^2 - \hat\sigma_{\theta'}^2| \leq 1024\nu_k\nu_l(\nu_k L_l\rho_\Gamma + \nu_l L_k\rho_\Omega). \tag{12}$$

Next, using Lemma 10 and Lemma 11 followed by a union bound over the $T_\Omega T_\Gamma$ center combinations gives us, with probability at least $1 - \delta$,

$$\max_{\substack{\omega' \in \{\omega_1,\ldots,\omega_{T_\Omega}\} \\ \gamma' \in \{\gamma_1,\ldots,\gamma_{T_\Gamma}\}}} |\hat\sigma_{\theta'}^2 - \sigma_{\theta'}^2| \leq 6144\nu_k^2\nu_l^2\sqrt{\frac{2}{n}\log\frac{2T_\Omega T_\Gamma}{\delta}} + \frac{4608\nu_k^2\nu_l^2}{n} \tag{13}$$

$$\leq \frac{2048\nu_k^2\nu_l^2}{\sqrt{n}}\left(3\sqrt{2\log\frac{2}{\delta} + 2D_\Omega\log\frac{4R_\Omega}{\rho_\Omega} + 2D_\Gamma\log\frac{4R_\Gamma}{\rho_\Gamma}} + \frac{9}{4\sqrt{n}}\right).$$

Finally, we combine Equations (11) to (13) to get

$$\sup_{\theta\in\Theta}|\hat\sigma_\theta^2 - \sigma_\theta^2| \leq \frac{2048\nu_k^2\nu_l^2}{\sqrt{n}}\left(3\sqrt{2\log\frac{2}{\delta} + 2D_\Omega\log\frac{4R_\Omega}{\rho_\Omega} + 2D_\Gamma\log\frac{4R_\Gamma}{\rho_\Gamma}} + \frac{9}{4\sqrt{n}} + \sqrt{n}\left(\frac{L_k}{\nu_k}\rho_\Omega + \frac{L_l}{\nu_l}\rho_\Gamma\right)\right).$$

Setting $\rho_\Omega = \rho_\Gamma = 1/\sqrt{n}$ gives us our desired uniform convergence bound.

$\square$

**Lemma 10.** *For any kernels $k$ and $l$ satisfying Assumption (A), with probability at least $1 - \delta$ we have*

$$|\hat\sigma^2 - \mathbb{E}\,\hat\sigma^2| \leq 6144\nu_k^2\nu_l^2\sqrt{\frac{2}{n}\log\frac{2}{\delta}}.$$

*Proof.* We apply McDiarmid's inequality to $\hat\sigma^2$. First, we show that the variance estimator satisfies bounded differences. For convenience, we denote $(i, j, q, r) \in \mathbf{i}_4^n$ simply as $(i, j, q, r)$, and $(i, j, q)\setminus k$ to be the set of 3-tuples drawn without replacement from $\mathbf{i}_3^n$ that exclude the number $k$. Recall that

$$\hat\sigma^2 = 16\left(\frac{1}{(n)_4(n-1)_3}\sum_{\substack{(i,j,q,r) \\ (b,c,d)\setminus i}} H_{ijqr}H_{ibcd} - \hat\eta^2\right).$$

Let $F$ denote the kernel tensor $H$ but with sample $(X_\ell, Y_\ell)$ replaced by $(X_\ell', Y_\ell')$ so that $F$ agrees with $H$ except at indices $\ell$, and let $\hat\eta'$ and $\hat\sigma'^2$ denote the HSIC and its variance estimators according to this updated sample set. The deviation is then

$$|\hat\sigma^2 - \hat\sigma'^2| \leq \frac{16}{(n)_4(n-1)_3}\sum_{\substack{(i,j,q,r) \\ (b,c,d)\setminus i}} |H_{ijqr}H_{ibcd} - F_{ijqr}F_{ibcd}| + 16|\hat\eta^2 - \hat\eta'^2|.$$

We bound the first term by noticing that $\Delta := H_{ijqr}H_{ibcd} - F_{ijqr}F_{ibcd}$ is zero when none of the indices $\{i,j,q,r,b,c,d\}$ is $\ell$. Let $S := \{(i,j,q,r,b,c,d) : (i,j,q,r) \in \mathbf{i}_4^n, (b,c,d) \in \mathbf{i}_3^n\backslash\{i\}, \ell \in \{i,j,q,r,b,c,d\}\}$ be the set of indices where $\Delta$ may be non-zero. By Assumption (A) we know that $|\Delta| \le 32\nu_k^2\nu_l^2$. Thus, we can bound the first term by

$$\frac{16}{(n)_4(n-1)_3}\sum_S |\Delta| = \frac{512\nu_k^2\nu_l^2}{(n)_4(n-1)_3}|S|$$

$$= \frac{512\nu_k^2\nu_l^2}{(n)_4(n-1)_3}\left[\underbrace{4(n-1)_3^2}_{\ell\in\{i,j,q,r\}} + \underbrace{3(n-1)(n-2)_2(n-1)_3}_{\ell\in\{b,c,d\}} - \underbrace{9(n-1)_3(n-2)_2}_{\ell\in\{j,q,r\}\ \text{and}\ \ell\in\{b,c,d\}}\right]$$

$$= 512\nu_k^2\nu_l^2\left(\frac{16}{n} - \frac{9}{n-1}\right)$$

$$\le \frac{8192\nu_k^2\nu_l^2}{n} \quad (\forall n > 1).$$

We can bound the second term using $|\hat{\eta}| \le 4\nu_k\nu_l$ (Assumption (A)) and the bounded difference result (10) from Proposition 8:

$$16|\hat{\eta}^2 - \hat{\eta}'^2| = 16|\hat{\eta} + \hat{\eta}'||\hat{\eta} - \hat{\eta}'| \le 16 \cdot 8\nu_k\nu_l \cdot \frac{32\nu_k\nu_l}{n} = \frac{4096\nu_k^2\nu_l^2}{n}.$$

Combining these two terms, the maximal bounded difference for $\hat{\sigma}^2$ is

$$|\hat{\sigma}^2 - \hat{\sigma}'^2| \le \frac{12288\nu_k^2\nu_l^2}{n}.$$

Finally, applying McDiarmid's inequality gives us, with probability at least $1 - \delta$,

$$|\hat{\sigma}^2 - \mathbb{E}\,\hat{\sigma}^2| \le 6144\nu_k^2\nu_l^2\sqrt{\frac{2}{n}\log\frac{2}{\delta}}.$$

$\square$

**Lemma 11.** *For any kernels $k$ and $l$ satisfying Assumption (A), the bias is bounded by*

$$|\mathbb{E}\,\hat{\sigma}^2 - \sigma^2| \le \frac{4608\nu_k^2\nu_l^2}{n}.$$

*Proof.* The expectation of the variance estimator is

$$\mathbb{E}\,\hat{\sigma}^2 = 16\left(\frac{1}{(n)_4(n-1)_3}\sum_{\substack{(i,j,q,r)\\(b,c,d)\backslash i}}\mathbb{E}[H_{ijqr}H_{ibcd}] - \frac{1}{(n)_4^2}\sum_{\substack{(i,j,q,r)\\(a,b,c,d)}}\mathbb{E}[H_{ijqr}H_{abcd}]\right).$$

First, we can break down the left-hand sum into only terms of $\mathbb{E}[H_{1234}H_{1567}]$ by considering the cases where $\{i,j,q,r,b,c,d\}$ are unique. Let $S = \{(i,j,q,r,b,c,d) : (i,j,q,r) \in \mathbf{i}_4^n, (b,c,d) \in \mathbf{i}_3^n\backslash\{i\}\}$ be the set of all possible indices of our left-hand sum. It follows that

$$\sum_S \mathbb{E}[H_{ijqr}H_{ibcd}] = \sum_{(i,j,q,r,b,c,d)\in\mathbf{i}_7^n}\mathbb{E}[H_{ijqr}H_{ibcd}] + \sum_{S\backslash\mathbf{i}_7^n}\mathbb{E}[H_{ijqr}H_{ibcd}].$$

If all indices are unique, then the expectation $\mathbb{E}[H_{ijqr}H_{ibcd}]$ is equivalent to $\mathbb{E}[H_{1234}H_{1567}]$; otherwise, we can bound the expectation by $16\nu_k^2\nu_l^2$ via Assumption (A). Thus, the bound on the left-hand sum is

$$\sum_{\substack{(i,j,q,r)\\(b,c,d)\backslash i}}\mathbb{E}[H_{ijqr}H_{ibcd}] \le (n)_7\,\mathbb{E}[H_{1234}H_{1567}] + \Big((n)_4(n-1)_3 - (n)_7\Big)16\nu_k^2\nu_l^2.$$

Similarly, we can break down the right-hand sum into only terms of $\mathbb{E}[H_{1234}H_{5678}]$. Let $R = \{(i,j,q,r,a,b,c,d) : (i,j,q,r) \in \mathbf{i}_4^n, (a,b,c,d) \in \mathbf{i}_4^n\}$ be the possible indices of our right-hand sum. We have that

$$\sum_{\substack{(i,j,q,r) \\ (a,b,c,d)}} \mathbb{E}[H_{ijqr}H_{abcd}] = \sum_{(i,j,q,r,a,b,c,d) \in \mathbf{i}_8^n} \mathbb{E}[H_{ijqr}H_{abcd}] + \sum_{R \setminus \mathbf{i}_8^n} \mathbb{E}[H_{ijqr}H_{abcd}]$$

$$\leq (n)_8 \, \mathbb{E}[H_{1234}H_{5678}] + \left((n)_4^2 - (n)_8\right) 16\nu_k^2 \nu_l^2.$$

Now, using these two results and Assumption (A), we can compute a bound on the desired bias of $\hat{\sigma}^2$:

$$|\mathbb{E}\,\hat{\sigma}^2 - \sigma^2| = 16 \left| \frac{1}{(n)_4(n-1)_3} \sum_{\substack{(i,j,q,r) \\ (b,c,d) \setminus i}} \mathbb{E}[H_{ijqr}H_{ibcd}] - \mathbb{E}[H_{1234}H_{1567}] - \frac{1}{(n)_4^2} \sum_{\substack{(i,j,q,r) \\ (a,b,c,d)}} \mathbb{E}[H_{ijqr}H_{abcd}] + \mathbb{E}[H_{1234}H_{5678}] \right|$$

$$\leq 16 \left| \left(1 - \frac{(n)_7}{(n)_4(n-1)_3}\right) \left(16\nu_k^2\nu_l^2 - \underbrace{\mathbb{E}[H_{1234}H_{1567}]}_{-16\nu_k^2\nu_l^2 \leq \cdot \leq 16\nu_k^2\nu_l^2}\right) + \left(1 - \frac{(n)_8}{(n)_4^2}\right) \left(\underbrace{\mathbb{E}[H_{1234}H_{5678}]}_{0 \leq \cdot \leq 16\nu_k^2\nu_l^2} - 16\nu_k^2\nu_l^2\right) \right|.$$

$$\leq 16 \left(1 - \frac{(n)_7}{(n)_4(n-1)_3}\right) 32\nu_k^2\nu_l^2 < 512\nu_k^2\nu_l^2 \cdot \frac{9}{n} \quad (\forall n \geq 4)$$

$$= \frac{4608\nu_k^2\nu_l^2}{n}. \qquad\qquad\qquad \Box$$

# C   ADDITIONAL EXPERIMENTS

## C.1   HIGH-DIMENSIONAL GAUSSIAN MIXTURE

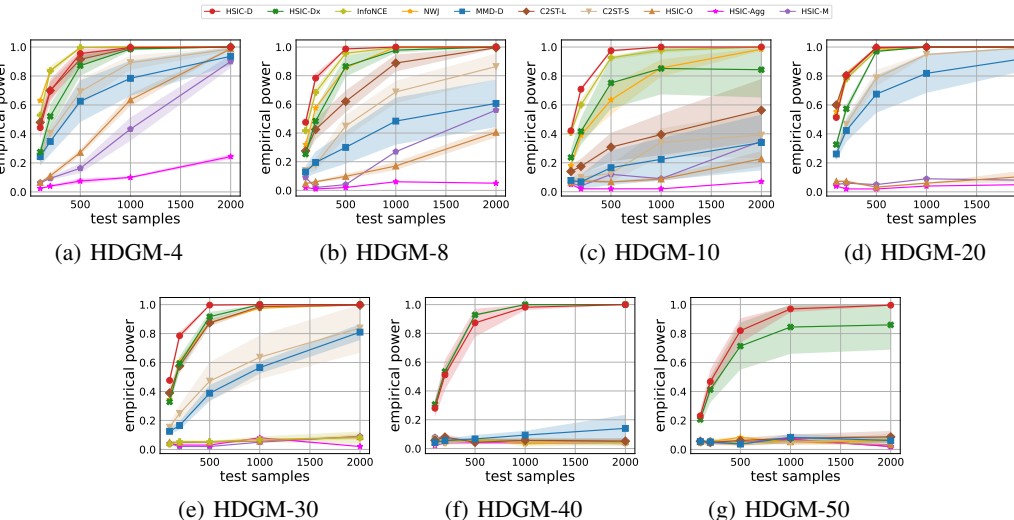

(a) HDGM-4     (b) HDGM-8     (c) HDGM-10     (d) HDGM-20

(e) HDGM-30     (f) HDGM-40     (g) HDGM-50

Figure 5: Empirical power vs test size for the HDGM problem at dimensions $d = \{2, 4, 5, 10, 15, 20\}$. The average test power is computed over 5 training instances and the shaded region covers one standard error from the mean.

## C.2   TYPE-I ERROR

Table 1 shows that the type-I error rates for our optimization-based tests are well-controlled.

| Method | HDGM-4 | HDGM-8 | HDGM-10 | HDGM-20 | HDGM-30 | HDGM-40 | HDGM-50 | Sinusoid | RatInABox |
|--------|--------|--------|---------|---------|---------|---------|---------|----------|-----------|
| HSIC-D | 0.043 | 0.043 | 0.050 | 0.050 | 0.062 | 0.057 | 0.052 | 0.050 | 0.048 |
| MMD-D | 0.048 | 0.055 | 0.040 | 0.053 | 0.048 | 0.048 | 0.055 | 0.054 | 0.050 |
| C2ST-L | 0.060 | 0.030 | 0.053 | 0.048 | 0.053 | 0.058 | 0.045 | 0.046 | 0.048 |
| InfoNCE | 0.046 | 0.046 | 0.046 | 0.054 | 0.044 | 0.050 | 0.048 | 0.048 | 0.045 |
| NWJ | 0.050 | 0.054 | 0.058 | 0.052 | 0.044 | 0.064 | 0.054 | 0.052 | 0.042 |

Table 1: Average type-I error rates under the null distribution over 400 tests. We use $m = 512$ samples.

## C.3   MMD VS. HSIC: SAMPLE PAIRINGS

One possible explanation why HSIC outperforms MMD in independence testing is that it utilizes paired samples more efficiently. Consider the mean of the product kernel under the null hypothesis, $\mathbb{E}[k(X, \tilde{X})l(Y, \tilde{Y})]$, which exists as a component of HSIC. The U-statistic estimator of this term is

$$\frac{1}{(m)_4} \sum_{(i,j,q,r) \in \mathbf{i_4^m}} k(X_i, X_j)l(Y_q, Y_r). \tag{14}$$

For MMD framed as an independence problem, we first permute the samples $(Y_i)_{i=1..m}$ by some permutation group $\sigma$ to simulate the null hypothesis, and then perform an MMD test between $(X_i, Y_i)_{i=1..m}$ and $(X_i, Y_{\sigma(i)})_{i=1..m}$. Under this approach, the mean product kernel term $\mathbb{E}[k(X, \tilde{X})l(Y, \tilde{Y})]$ is estimated by

$$\frac{1}{(m)_2} \sum_{(i,j) \in \mathbf{i_2^m}} k(X_i, X_j)l(Y_{\sigma(i)}, Y_{\sigma(j)}).$$

This estimator is biased and seems to suffer from high variance, which we detail in the following Appendix C.4. Additionally, we expect that as we consider more permutation groups, i.e., more samples from the null distribution, the biased MMD estimator should more closely approximate the biased HSIC estimator. To see this, consider the collection of permutation groups $\{\sigma_q\}_{q=1..m}$ satisfying $\sigma_m(i) = i$ and $\sigma_j(i) = \sigma_{j+1}(i+1)$ so that each successive permutation group is a shifting of its elements to the right by one position. Then, the samples $(X_i, Y_{\sigma_q(i)})_{i\in[m], q\in[m]}$ consist of all possible $m^2$ pairings of $(X_i, Y_j)_{i\in[m]j\in[m]}$. For an MMD test between $(X_i, Y_i)_{i=1..m}$ and all pairings $(X_i, Y_{\sigma_q(i)})_{i\in[m], q\in[m]}$, a biased estimator for the mean product kernel term under $\mathfrak{H}_0$ is

$$\frac{1}{m^4} \sum_{i,j,q,r} k(X_i, X_j) l(Y_{\sigma_q(i)}, Y_{\sigma_r(j)})$$

which is equivalent to the V-statistic estimator used in HSIC. We test this equivalence by plotting the empirical power of MMD-based independence tests using varying numbers of permutation groups $\{\sigma_q\}_{q=1..G}$ from $G = 1$ to $G = m$ in Figure 6. We see that as we consider more permutation groups, which corresponds to using more pairs $(X_i, Y_j)$ from the null distribution, the MMD test power seems to converge to the HSIC test power.

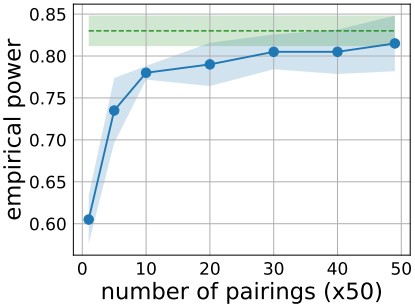

Figure 6: We perform an MMD-based independence test (blue) on $m = 50$ samples from the Sinusoid problem with frequency $\ell = 1$. We use Gaussian kernels with bandwidth 2 for both $k$ and $l$. We vary the number of pairings $(X_i, Y_j)$ used for simulating the null hypothesis from $m$ to $m(m-1)$. The green line corresponds to an HSIC-based independence test under identical settings. The shaded region indicates one standard deviation from the mean over 5 tests.

### C.4  MMD vs. HSIC: Asymptotic Variance

We show estimates of the asymptotic variance of HSIC and MMD along a training trajectory in Figure 7. For MMD we consider both a single shuffling of the data (MMD-full), as well as a split shuffling (MMD-split) where we use half the data for our joint distribution sample, and the other half to permute for our product-of-marginals sample.

We note that the initial variance of MMD-split is substantially higher than that of MMD-full, which is much higher than the variance of HSIC. We hypothesize this makes the MMD-based objective harder to optimize, and gives a possible explanation for why MMD-split performs worse than MMD-full. MMD-split/full also exhibit greater final variances, particularly at larger batch sizes.

### C.5  Optimizing J vs. HSIC

We examine the trade-off between optimizing the approximate test power $J$ versus just the test statistic HSIC. The results on power versus test size are show in Figure 8. Optimizing our proposed objective $J$ significantly outperforms optimizing HSIC for all problems.

### C.6  Training Times

We compare the training times of all methods in Table 2. Our method is comparable in speed to C2ST and MMD, and vastly faster than the variational mutual information bound methods InfoNCE and

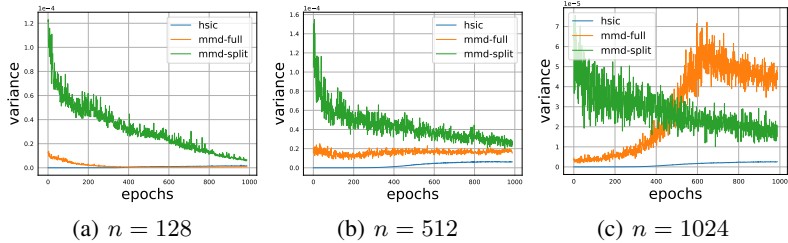

(a) $n = 128$        (b) $n = 512$        (c) $n = 1024$

Figure 7: Estimates of the asymptotic variance of HSIC (blue), MMD with a single permutation (orange), and MMD with a split permutation (green) along a training trajectory for HDGM-10 at sample sizes n=128 (a), n=512 (b), and n=1024 (c).

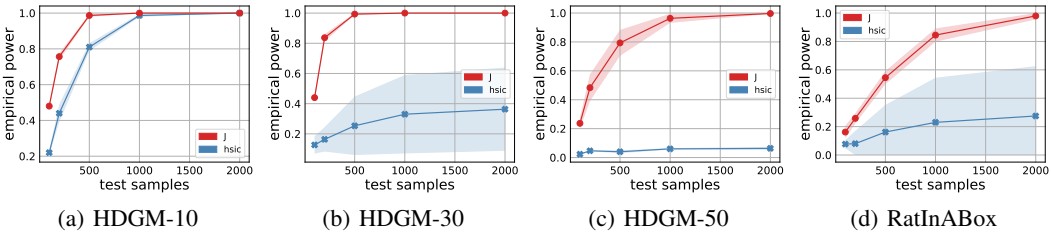

(a) HDGM-10     (b) HDGM-30     (c) HDGM-50     (d) RatInABox

Figure 8: Test power using deep kernels optimized for the approximate asymptotic test power $J$ (red) versus optimizing just the test statistic HSIC (blue).

NWJ since both those methods require function evaluations for *every* possible pair of samples at each training step.

Table 2: Approximate training times for the HDGM-30 problem over 1,000 epochs on a GeForce RTX 4060. Times are rounded to the nearest minute

|  | Training Time |
|---|---|
| HSIC | 41 m |
| C2ST | 34 m |
| MMD | 44 m |
| InfoNCE | 8 h 38 m |
| NWJ | 8 h 42 m |

# D EXPERIMENT DETAILS

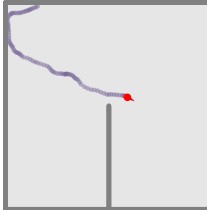

Figure 9: RatInABox simulation environment. The red dot is the current position of the rat and the purple circles indicate the past trajectory over 5 seconds. The box is designed to have only a single protruding wall.

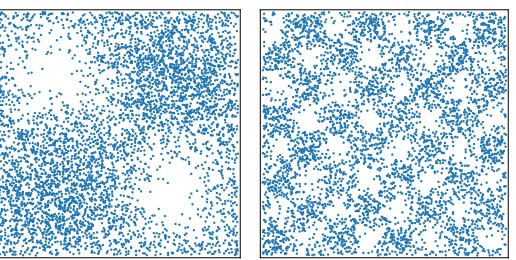

Figure 10: Samples drawn from the sinusoidal problem with frequency $\ell = 1$ (left) and $\ell = 4$ (right). We consider the latter frequency in our experiments.

## D.1 TRAINING & TEST DETAILS

We design the featurizers $\phi_\omega$ and $\phi_\kappa$ of our deep kernels $k_\omega$ and $l_\kappa$ to be neural networks with ReLU activations. We avoid using normalization as it may affect test power. Moreover, we make the Gaussian bandwidth of both $k_\omega$ and $l_\kappa$ a learnable parameter, as well as the smoothing rate $\epsilon$. To make comparisons as fair as possible, we use similar neural network architectures for each deep learning based method. In general, we let the featurizer of HSIC-D and MMD-D be identical up to a concatenation layer which concatenates $X$ and $Y$ to frame the problem as a two-sample test. We construct the C2ST-S/L classifier as the MMD-D featurizer plus a linear layer classification head with scalar output, and we let both C2ST-S/L and InfoNCE use identical architectures for the classifier and critic. Detailed descriptions of each architecture are demonstrated in the following subsections.

All optimization-based methods (HSIC-D/Dx/O, MMD-D, C2ST-S/L, InfoNCE) are first trained on an identical split of the data, and then tested on the remaining split. In contrast, HSIC-M selects the median bandwidth based on the entire dataset, and is evaluated on the test set. We train our models using the AdamW optimizer with a learning rate of 1e-4 over 1,000 epochs for HDGM and RatInABox and 10,000 epochs for Sinusoid. We use a batch size of 512. All methods are implemented in PyTorch and trained on a NVIDIA A100SXM4 GPU.

Once learned, each methods' empirical power is evaluated on 100 test sets $(S_Z^{te_1}, ..., S_Z^{te_{100}})$. Each test set contains $m$ test samples $S_Z^{te_i} = (Z_1^{te_i}, ..., Z_N^{te_i})$, which are then used to compute the average rejection rate under the null via a permutation test. We use 500 permutations for each test and with a predetermined type-I error rate of 0.05.

## D.2 ARCHITECTURES

In all experiments we consider deep kernels with Gaussian feature and smoothing kernels $\kappa$ and $q$, where each bandwidth is a trainable parameter randomly initialized around $1.0$. We let the smoothing weight $\epsilon$ also be a learnable parameter initialized to $0.01$. No batch normalization is used and all hidden layers use ReLU activations. Dataset-specific designs are elaborated below.

**High-dimensional Gaussian mixture**. We use a feed-forward network for our deep kernel featurizer with latent dimensions $2d, 3d$, and $2d$. Details of each model is given in Table 3.

**Sinusoid**. The deep kernel featurizer is taken to be a feed-forward network with widths 1x8x12x8. C2st, infoNCE, and NWJ use a similar architecture –one with widths 2x8x12x8x1– which includes an additional scalar output layer.

**RatInABox**. We use a feed-forward featurizer with details given in Table 4. Unlike the previous two problems, the sample spaces $\mathcal{X}$ and $\mathcal{Y}$ are not equivalent, and so the deep featurizers for $k$ and $l$ have different architectures.

| dataset | model | input | featurizer |
|---|---|---|---|
| HDGM-4 | HSIC-D | X or Y | $[\,2 \to 4 \to 6 \to 4\,]$ |
| | MMD-D | [X, Y] | $[\,4 \to 8 \to 12 \to 8\,]$ |
| | C2ST-S/L | [X, Y] | $[\,4 \to 8 \to 12 \to 8 \to 1\,]$ |
| HDGM-8 | HSIC-D | X or Y | $[\,4 \to 8 \to 12 \to 8\,]$ |
| | MMD-D | [X, Y] | $[\,8 \to 16 \to 24 \to 16\,]$ |
| | C2ST-S/L | [X, Y] | $[\,8 \to 16 \to 24 \to 16 \to 1\,]$ |
| HDGM-10 | HSIC-D | X or Y | $[\,5 \to 10 \to 15 \to 10\,]$ |
| | MMD-D | [X, Y] | $[\,10 \to 20 \to 30 \to 20\,]$ |
| | C2ST-S/L | [X, Y] | $[\,10 \to 20 \to 30 \to 20 \to 1\,]$ |
| HDGM-20 | HSIC-D | X or Y | $[\,10 \to 20 \to 30 \to 20\,]$ |
| | MMD-D | [X, Y] | $[\,20 \to 40 \to 60 \to 40\,]$ |
| | C2ST-S/L | [X, Y] | $[\,20 \to 40 \to 60 \to 40 \to 1\,]$ |

| dataset | model | input | featurizer |
|---|---|---|---|
| HDGM-30 | HSIC-D | X or Y | $[\,15 \to 30 \to 45 \to 30\,]$ |
| | MMD-D | [X, Y] | $[\,30 \to 60 \to 90 \to 60\,]$ |
| | C2ST-S/L | [X, Y] | $[\,30 \to 60 \to 90 \to 60 \to 1\,]$ |
| HDGM-40 | HSIC-D | X or Y | $[\,20 \to 40 \to 60 \to 40\,]$ |
| | MMD-D | [X, Y] | $[\,40 \to 80 \to 120 \to 80\,]$ |
| | C2ST-S/L | [X, Y] | $[\,40 \to 80 \to 120 \to 80 \to 1\,]$ |
| HDGM-50 | HSIC-D | X or Y | $[\,25 \to 50 \to 75 \to 50\,]$ |
| | MMD-D | [X, Y] | $[\,50 \to 100 \to 150 \to 100\,]$ |
| | C2ST-S/L | [X, Y] | $[\,50 \to 100 \to 150 \to 100 \to 1\,]$ |

Table 3: Featurizer architectures used in deep kernels for HSIC-D, MMD-D, and classifier architecture used for C2ST-S/L on the HDGM problem. Brackets denote a sequence of linear layers with corresponding input and output features.

| method | input | network |
|---|---|---|
| HSIC-D | X | $[\ 8 \to 32 \to 64 \to 32\ ]$ |
| | Y | $[\ 2 \to 4 \to 8 \to 4\ ]$ |
| MMD-D | [X, Y] | $[\ 10 \to 32 \to 64 \to 32\ ]$ |
| C2ST-S/L | [X, Y] | $[\ 10 \to 32 \to 64 \to 32 \to 1\ ]$ |

Table 4: Featurizer architectures used in deep kernels for HSIC-D, MMD-D, and classifier architecture used for C2ST-S/L on the RatInABox problem. Brackets denote a sequence of linear layers with corresponding input and output features.

# E    REBUTTAL MATERIAL

We'll integrate this into the rest of the paper in a future revision.

## E.1    WINE DATASET

● **Wine Quality** (Paulo et al., 2009). Details physicochemical properties (e.g., sugar, pH, chlorides) of different types of red and white wines and their perceived quality as an integer value from 1 to 10. We test for dependency between residual sugar levels and quality.

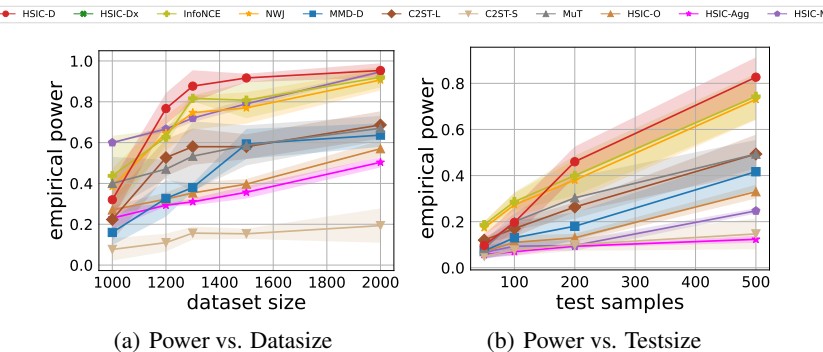

(a) Power vs. Datasize      (b) Power vs. Testsize

Figure 11: (a) Empirical power vs dataset size. We train optimization-based methods with a 3:2 train-test split across all dataset sizes. HSIC-Agg/M use the entire dataset for testing. (b) Empirical power vs test size. All optimization-based models are first trained on 2000 samples with a 3:2 train-test split, then evaluated at test sizes of $m = \{50, 100, 200, 500\}$. The shaded region covers one standard deviation over 3 training runs

## E.2    MUT TEST

This section evaluates tests based on maximizing the signal-to-noise ratio $(T - T_0)/\sigma_{\mathfrak{H}_1}$ of the $\hat{T}$ statistic, as discussed in the beginning of Section 5. We estimate this quantity on a batch of $m$ samples with

$$\frac{\hat{T} - \hat{T}_0}{\hat{\sigma}_{\mathfrak{H}_1}} = \frac{\frac{1}{m}\sum_{i=1}^m f(X_i, Y_i) - \frac{1}{m^2}\sum_{i=1}^m\sum_{j=1}^m f(X_i, Y_j)}{\sqrt{\frac{1}{m}\sum_{i=1}^m \left(f(X_i, Y_i) - \frac{1}{m}\sum_{j=1}^m f(X_j, Y_j)\right)^2 + \lambda}}$$

for a small positive $\lambda$. To choose an arbitrary name, we call these tests MuT for now.

### E.2.1    EXPERIMENTAL RESULTS

Results on the HDGM, Sinusoid and RatInABox problems are shown below.

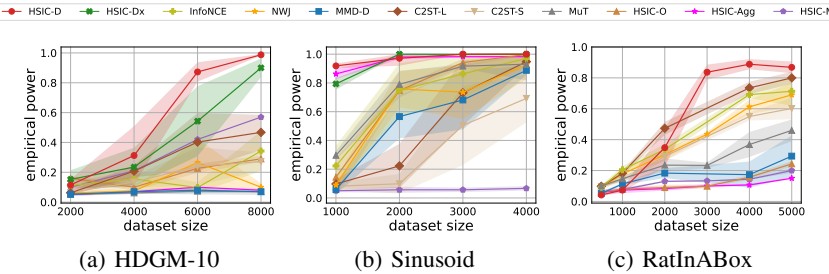

(a) HDGM-10      (b) Sinusoid      (c) RatInABox

Figure 12: Empirical power vs dataset size for (a) HDGM-10, (b) Sinusoid, and (c) RatInABox. The gray line corresponds to optimizing the MuT SNR. Note that in (a), MuT is hidden behind other baselines (with trivial 0.05 power).

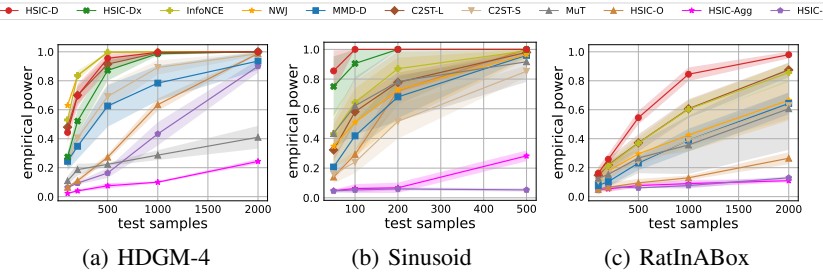

(a) HDGM-4      (b) Sinusoid      (c) RatInABox

Figure 13: Empirical power vs test size for (a) HDGM-4, (b) Sinusoid, and (c) RatInABox. The gray line corresponds to optimizing the MuT SNR.

### E.2.2 WHY DOESN'T IT WORK BETTER?

Strikingly, optimizing the estimated asymptotic power of the test based on the critic doesn't seem to do better than choosing the critic based on the mutual information bounds InfoNCE or NWJ. Why is this? First, let's compare MuT to InfoNCE on RatInABox more closely in Table 5.

| | MuT | | | InfoNCE | | |
|---|---|---|---|---|---|---|
| **Run** | **Power@128** | **Power@2000** | **SNR** | **Power@128** | **Power@2000** | **SNR** |
| 1 | 0.09 ± 0.02 | 0.39 ± 0.03 | 0.03 ± 0.02 | 0.13 ± 0.02 | 0.75 ± 0.04 | 0.007 ± 0.003 |
| 2 | 0.09 ± 0.03 | 0.30 ± 0.02 | 0.02 ± 0.02 | 0.15 ± 0.03 | 0.81 ± 0.03 | 0.011 ± 0.005 |
| 3 | 0.06 ± 0.02 | 0.11 ± 0.03 | 0.02 ± 0.02 | 0.13 ± 0.04 | 0.66 ± 0.03 | 0.010 ± 0.005 |

Table 5: SNR values of MuT and InfoNCE tests over three runs on the RatInABox problem. We use 3000 training samples; each value is the mean and standard deviation of 100 tests for the given trained critic.

The MuT tests do indeed obtain larger average SNR values (by a factor of two or three), but have notably lower empirical powers, contradicting our argument about the asymptotic power. For both models, we empirically observe that the ratio $\sigma_{\mathfrak{H}_0}/\sigma_{\mathfrak{H}_1} \approx 1$ (see Figure 14); thus the asymptotic power should be roughly $\Phi\left(\sqrt{m}\,\mathrm{SNR} - 1 \cdot \Phi^{-1}(1-\alpha)\right)$. Taking SNR 0.025 and $\alpha = 0.05$ gives a predicted power of approximately 0.09 for $m = 128$ and 0.30 for $m = 2000$; this agrees fairly closely with the empirical power obtained by MuT. The asymptotics are thus apparently explaining this test well.

With SNR 0.01, however, the predicted power of these InfoNCE tests would be 0.06 and 0.11; far below the actual observed power of about 0.13 and 0.75. What happened?

Figure 15 shows that the particular value of the test statistic and the rejection threshold are highly dependent on one another. (Each test corresponds to separately sampled test sets, with the same critic function.) There is a substantial 45% correlation between the two for MuT, but for InfoNCE

the correlation is 99%. While the permutation tests do achieve the appropriate level, this coupling between the test statistic and the test threshold is not accounted for in our asymptotic analysis, explaining InfoNCE's strong departure from that regime. We do not yet understand why InfoNCE exhibits *such* strong coupling here; this is a very interesting area for future work.

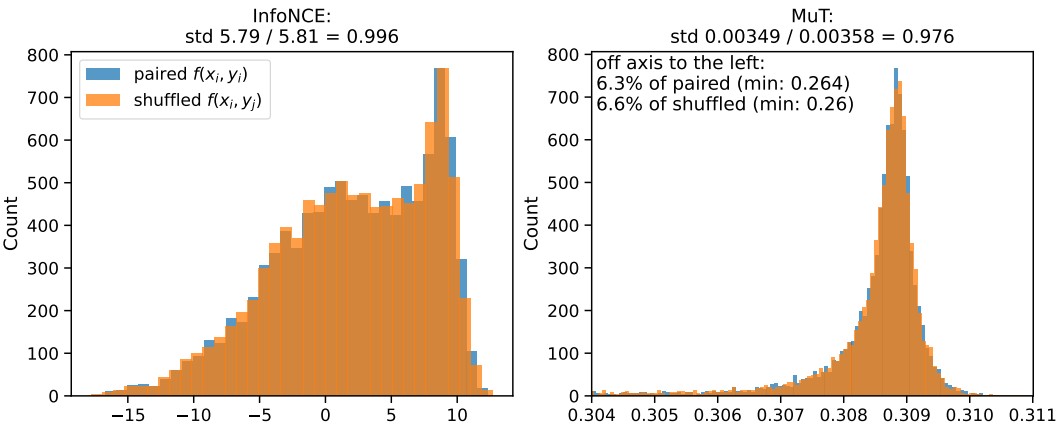

Figure 14: Histograms of $f(x_i, y_i)$ and $f(x_i, y_j)$ across 10,000 samples (with shuffled points based on a single permutation), for run 1 on RatInABox from Table 5.

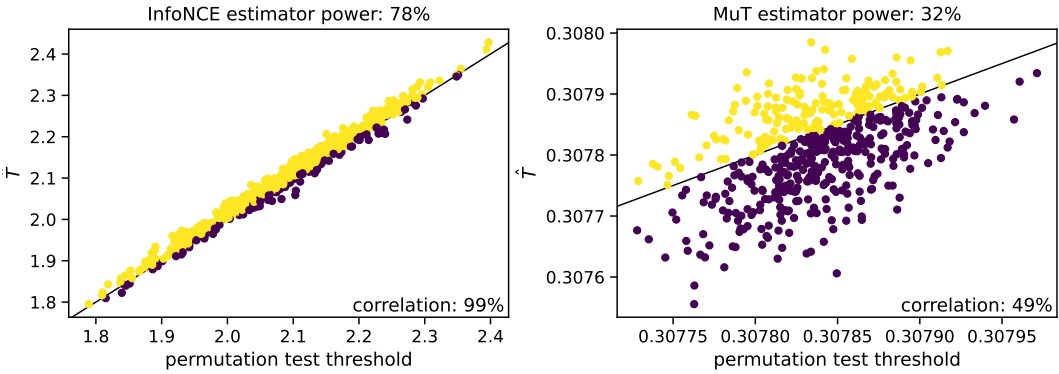

Figure 15: The estimated $\hat{T}$ and the rejection threshold obtained by permutation testing with level $\alpha$, for 500 sample sets of run 1 on RatInABox from Table 5. The line corresponding to the rejection boundary is marked, and rejecting points are colored yellow.

