# OpenReview forum: "Learning Representations for Independence Testing"
_ICLR.cc/2025/Conference — Submitted to ICLR 2025_

### Official Review · Reviewer_QkMU · 2024-10-21

**Soundness:** 3
**Presentation:** 3
**Contribution:** 3
**Rating:** 6
**Confidence:** 2

**Summary:**

The paper introduces new methods to detect dependencies between random variables in high-dimensional data by utilizing variational estimators of MI  and the HSIC. It aims to improve the power of independence tests by leveraging deep learning techniques to learn effective representations of the data, making the dependencies between variables more explicit. The paper also highlights a connection between MI-based tests and HSIC-based tests, showing that they are closely related. The paper introduces the concept of maximizing test power using learned representations, specifically focusing on HSIC-based tests, which are generally found to outperform MI-based ones. The experiments are abundant and support the claims.

**Strengths:**

The paper is well-organized and clearly presented. The authors clearly explain their methods and provide intuitive connections between the variational bounds and kernel-based methods.

Theoretically, the paper is the first to construct a valid independence test based on the variational mutual information estimator in high dimensions. The paper provides theoretical insights into the connection between HSIC and MI tests and shows how to maximize the test power for independence testing using learned representations.

Although the experiments were all conducted on synthetic datasets, the experimental validation is abundant with various settings. The empirical results show that the performance of proposed HSIC-based tests is nontrivial especially in high dimensions.

**Weaknesses:**

One limitation could be the lack of evaluation on real-world datasets despite the authors already presenting three quite informative synthetic datasets. Since the presented tests only focus on testing independence, experiments of causal discovery may not be suitable. But I think testing a pair of variables is still executable. Experiments on any real-world datasets are appreciated, e.g., https://www.cmu.edu/dietrich/causality/projects/causal_learn_benchmarks/. This is just an example, the author does not need to be constrained by this and can use any real-world dataset. If there are any reasons why experiments on real-world datasets may not be feasible or practical, I would be very interested in hearing them and would welcome the explanation with an open mind.

Another limitation, which is also discussed by the authors, is the scope only focusing on the independence test, which is considerably easier compared with the conditional independence test.

A minor piece of advice: A paragraph or several sentences to highlight the difference between the traditional independence test and the independence test in high dimensions will be appreciated and suitable for readers from different levels.

**Questions:**

- What is the definition of K in Line 130?
- You mention future work on extending this approach to conditional independence testing and causal discovery. Can you provide more details on what specific challenges you foresee in adapting the current method for conditional independence?

---

> ### Author Response · Authors · 2024-11-22
>
> Thank you for your constructive feedback and your review of our paper.
>
>
> > One limitation could be the lack of evaluation on real-world datasets [...]
>
> Thanks for bringing these datasets to our attention, and we appreciate this open-minded comment.
>
> In general, the challenge is that many of the existing real-world datasets are just "too simple," because they are designed to be approachable for classification or regression tasks. If a classifier or regressor gets any non-trivial performance, then the variables are clearly dependent (this is the basis of C2STs). On many of these datasets, median heuristic-based kernel tests achieve perfect power with only a few hundred samples. Our optimization-based methods, on the other hand, shine in problems with *subtle* dependencies, for which these simpler kernels are barely able to detect any dependence at all – not the kinds of datasets which tend to be available in machine learning repositories. We believe that such questions *do* arise in real scientific and engineering tasks, but lacking the tools to address them, people usually don't think to ask or publicize such problems.
>
> One real dataset where dependence is present but subtle is the Wine Quality dataset [Paulo et al., 2009], which details physicochemical properties (e.g., sugar, pH, chlorides) of different types of red and white wines, along with their perceived quality as an integer value from 1 to 10. We test dependency between residual sugar levels and quality, and plot our results in Figure 11 (Appendix D.3.1); our method outperforms all competitors for a range of dataset sizes, although the median heuristic is better at very small sizes and for large enough datasets all methods work well.
>
> We would be happy to hear any suggestions of other real datasets that we should try (without taking too much domain-knowledge effort for one application of an ICLR paper).
>
> > Another limitation, which is also discussed by the authors, is the scope only focusing on the independence test, which is considerably easier compared with the conditional independence test. Can you provide more details on what specific challenges you foresee in adapting the current method for conditional independence?
>
> Thanks for your comment. This is true; conditional independence (unless there are many samples per condition) is a more difficult problem and requires very different techniques, because we need to find the relationship between the condition variable and interested variables. The fundamental idea of optimizing the asymptotic test power could apply to conditional testing approaches like KCI [Zhang et al., 2011] and CIRCE [Pogodin et al. 2023], but the performance might largely depend on how well we model the relation between the conditioning variable and interested variables, which brings many challenges of its own beyond the setting of marginal independence. We will discuss this relationship in our revision.
>
>
>
> > A minor piece of advice: A paragraph or several sentences to highlight the difference between the traditional independence test and the independence test in high dimensions will be appreciated and suitable for readers from different levels.
>
> Thank you for your suggestion. We will expand on this difference in our revision.
>
>
> > What is the definition of K in Line 130?
>
> Thanks for bringing this to our attention. K is basically the test/batch size; we will clarify this.

---

> > ### Comment · Reviewer_QkMU · 2024-11-26
> >
> > I appreciate the authors for their dedicated effort in the rebuttal. My concerns are addressed. I will keep my score.

---

> > > ### Author Response · Authors · 2024-11-30
> > >
> > > Dear Reviewer QkMU,
> > >
> > > It is glad to hear that your concerns have been addressed well. Thanks for your support!
> > >
> > > Best regards,
> > >
> > > Authors of Submission 11733

---

### Official Review · Reviewer_WLby · 2024-10-31

**Soundness:** 3
**Presentation:** 2
**Contribution:** 3
**Rating:** 5
**Confidence:** 3

**Summary:**

The authors introduce two methods for independence testing, which are based on known methodologies. Two approaches are explored:
1. Tests based on variational mutual information estimators, such as InfoNCE,
2. The HSIC that aims to maximize test power in high-dimensional data.
By recasting the corresponding test as optimization problems over neural nets, the authors demonstrate that HSIC-based tests outperform MI-based ones, particularly for complex dependencies.

**Strengths:**

1. I like the idea of deriving an optimizer for independence testing directly from the hypothesis test error. This makes its motivation more explicit, rather than information theoretic methods which can be thought of as 'indirect', or inducing independence as a byproduct. The authors directly demonstrate how variational mi bounds are not aiming for P_e minimization.
2. The authors present formal guarantees of the proposed method and connect it to variational bounds of mutual information, which are highly popular methods that are widespread in the community.
3. When discussing the expression in neural estimation that is effectively required for independence testing - I liked the permutation invariance interpretation. (A note for the authors - this can be also seen from the proof of the DV formula)

**Weaknesses:**

1. I feel that the paper could benefit from better representation, which makes it harder for the reader to understand theciontributions in a bird's view, e.g., some crucial parts are migrated to the appendix, such as the assumptions of the main theorem of the paper, while some parts do not benefit the presentation in my opinion, such as some of the very specific and technical details in Section 5.
2. As the authors utilize 'heavy' machinery such as NN optimization, I would expect at least one experimental results on real world high dimensional data.
3. Literature review: Independence testing has been previously explored in the form of optimization of families of functions. Specifically, classical methods such as canonical correlation analysis (Hoteling 67) and its many extensions tackle the problem of independence testing. In an information theoretic setting there are the Hirschfeld-Gebelein-Renyi coefficient. More recently, max-sliced mutual information (Tsur 24), which was shown useful for independence testing and was connected to constrained variational mutual information estimation.
4. Section 2 lacks a bottom line, and currently feels like it 'stops in the middle'
5. I am not sure the citation formal is valid.

Technical comments:
1. undefined acronyms - InfoNCE, NWJ,DV
2. line 101 - 'method method'
3. line 136 - 'feature'-'features'
4. line 147 - 'by by'

**Questions:**

1. Can the authors comment on the tightness of the bound presented in proposition 3 (beyond the asymptotics)?
2. Does the MMD in line 255 depend on the function $f$ or function family $\cal{H}$? it is not clear from the notation, while I am pretty sure it does.
3. Could the authors discuss the assumptions of Theorem 4? I would also expect a short discussion on their 'reasonability' i.e., which distributions are captued under such assumptions?
4. Theorem discusses the implication of a unique maximizer. Do we know when such a maximizer exists?
5. Did the authos perform any ablation on the method? It would be specifically interesting to see if the networks size should indeed be linear in the input dimension.
5. The presentation would benefit from clarifying what is K in the InfoNCE notation.
6. "Gaussian kernel with unit bandwidth ... this scheme can perform extremely poorly" - the intro would benefit from a short discussion on such cases, as they form some of the motivation for this work.

---

> ### Author Response · Authors · 2024-11-22
>
> Thank you for your constructive feedback and your review of our paper.
>
> > (A note for the authors - this can be also seen from the proof of the DV formula)
>
> That's quite interesting, thanks for pointing this out! We'll add a footnote about this in the revision.
>
>
> > I feel that the paper could benefit from better representation, which makes it harder for the reader to understand the contributions in a bird's view, e.g., some crucial parts are migrated to the appendix, such as the assumptions of the main theorem of the paper, while some parts do not benefit the presentation in my opinion, such as some of the very specific and technical details in Section 5.
>
> Thanks for bringing this to our attention. Certain parts of the theory, particularly the assumptions, were moved to the appendix in the process of editing our paper mainly to fit in the page limits. We also believe that the assumptions are relatively benign, and hold for many problems and the particular deep kernels we use in practice; see Propositions 7 and 9 of [Liu et al., 2020]. We agree that we should at least discuss this in the main body; we will add at least some discussion, and hopefully statements of the assumptions if we can make them fit, to the main body in revision.
>
>
> > As the authors utilize 'heavy' machinery such as NN optimization, I would expect at least one experimental results on real world high dimensional data.
>
> Our experiments are indeed synthetic, but we want to emphasize that RatInABox was designed by neuroscientists as a potential replacement for running real neural experiments with live animals; it was designed to be as realistic as possible. Thus, while synthetic, we think it is not entirely not "real world data." If we recorded real rats running around a maze for several months, and applied standard neural processing pipelines, the dataset should look quite similar (at the cost of grad students' time and rats' lives).
>
> In general, one major challenge with real-world datasets for independence testing is that many of the easily usable real-world datasets are just "too simple," because they are designed to be approachable for classification or regression tasks. If a classifier or regressor gets any non-trivial performance, then the variables are clearly dependent (this is the basis of C2STs). On many of these datasets, median heuristic-based kernel tests achieve perfect power with only a few hundred samples. Our optimization-based methods, on the other hand, shine in problems with *subtle* dependencies, for which these simpler kernels are barely able to detect any dependence at all – not the kinds of datasets which tend to be available in machine learning repositories. We believe that such questions *do* arise in real scientific and engineering tasks, but lacking the tools to address them, people usually don't think to ask or publicize such problems.
>
> One real dataset where dependence is present but subtle is the Wine Quality dataset [Paulo et al., 2009], which details physicochemical properties (e.g., sugar, pH, chlorides) of different types of red and white wines, along with their perceived quality as an integer value from 1 to 10. We test dependency between residual sugar levels and quality, and plot our results in Figure 11 (Appendix E.1); our method outperforms all competitors for a range of dataset sizes, although the median heuristic is better at very small sizes and for large enough datasets all methods work well.
>
> We would be happy to hear any suggestions of other real datasets that we should try (without taking too much domain-knowledge effort for one application of an ICLR paper).
>
> > Literature review: Independence testing has been previously explored in the form of optimization of families of functions. Specifically, classical methods such as canonical correlation analysis (Hoteling 67) and its many extensions tackle the problem of independence testing. In an information theoretic setting there are the Hirschfeld-Gebelein-Renyi coefficient. More recently, max-sliced mutual information (Tsur 24), which was shown useful for independence testing and was connected to constrained variational mutual information estimation.
>
> Thanks for highlighting these works, some of which we were unaware of. We'll add a discussion of them to the related work section, but as far as we can tell, none seem to address our fundamental problem of learning an appropriate representation to test with. In particular, when optimizing over RKHS functions, HGR is closely related to a normalized version of HSIC; choosing the RKHS to use becomes very similar to the problem of choosing the RKHS to use in an HSIC test.
>
> The max-sliced mutual information is quite relevant, but the form of representation they consider (slicing) is extremely limited; we will discuss this further in revision.

---

> > ### Author Response · Authors · 2024-11-22
> >
> > > Section 2 lacks a bottom line, and currently feels like it 'stops in the middle'
> >
> > We will consider revising this section to make things flow more naturally. Thank you for bringing this to our attention.
> >
> >
> >
> >
> > > Can the authors comment on the tightness of the bound presented in proposition 3 (beyond the asymptotics)?
> >
> > This bound has two steps.
> >
> > The tightness of the HSIC to TV bound depends on how smooth in the RKHS the difference between the two distributions is. The bound is tight when there is an RKHS function which is a perfect indicator of which distribution is more likely at the present point.
> >
> > For the tightness of bounding the TV by the KL; see [Canonne, 2023]. The left bound is slightly tighter than the right for small values of the mutual information, while the right-hand bound is much tighter for large values. (The left-hand bound can never produce a lower bound greater than 1, unlike the right-hand one.)
> >
> >
> >
> > > Does the MMD in line 255 depend on the function or function family? it is not clear from the notation, while I am pretty sure it does.
> >
> > Yes, it depends on the function $f$, which determines the kernel used in the MMD. Thanks for pointing this out; we have changed the notation to $\text{MMD}_f$ to denote this dependency.
> >
> >
> >
> > > Could the authors discuss the assumptions of Theorem 4? I would also expect a short discussion on their 'reasonability' i.e., which distributions are captued under such assumptions?
> >
> > Theorem 4 assumes bounded kernels satisfying a certain form of Lipshitz continuity. We believe that the assumptions are relatively benign, and hold for many problems and the particular deep kernels we use in practice; see Propositions 7 and 9 of [Liu et al., 2020]. We agree that we should discuss this in the main body; we will add at least some discussion, and hopefully statements of the assumptions if we can make them fit, to the main body in revision.
> >
> >
> >
> >
> > > Theorem discusses the implication of a unique maximizer. Do we know when such a maximizer exists?
> >
> > Existence of a maximizer depends on the distribution/problem and kernel parameterization; with kernels defined by deep networks, this will generally not be true due to symmetries in the parameterization. That said, even if there is no unique maximizer to recover, the main point of our theorem is that the kernel we find will perform almost as well as any kernel can.
> >
> > > Did the authors perform any ablation on the method? It would be specifically interesting to see if the networks size should indeed be linear in the input dimension.
> >
> > The network architecture and size was just an arbitrary choice that made sense for these types of problems. If we instead worked with something like image data we could use CNN-based architectures (while still satisfying the assumptions of Theorem 4). The network size also does not have to be linear in the input dimension, but again made sense here. More importantly, we emphasize that the choice of architecture was made consistent across all deep learning based approaches (HSIC-D/C2ST/MMD/InfoNCE/NWJ) so that the comparisons are as fair as possible.
> >
> > As for ablations, HSIC-O vs HSIC-D is an ablation justifying the use of deep kernels over Gaussians with a trainable bandwidth. HSIC-Dx vs HSIC-D justifies simutaneously optimizing over kernels for both $X$ and $Y$ rather than a single shared kernel. Appendix C.5 is an ablation justifying optimizing $J$ (the HSIC SNR) over just optimizing the HSIC statistic.
> >
> > > The presentation would benefit from clarifying what is K in the InfoNCE notation.
> >
> > Thanks for bringing this to our attention. $K$ is basically the test/batch size; we will clarify this.
> >
> >
> >
> > > "Gaussian kernel with unit bandwidth ... this scheme can perform extremely poorly" - the intro would benefit from a short discussion on such cases, as they form some of the motivation for this work.
> >
> > Thanks for bringing this to our attention. We will add some more discussion on this.

---

> ### Author Response · Authors · 2024-11-28
> **Reminder - Discussion Stage Closing Soon - 28 November**
>
> Dear Reviewer WLby,
>
> Thank you for taking the time and effort to review our manuscript.
>
> We have carefully addressed all your comments and prepared detailed responses. Could you kindly review them at your earliest convenience?
>
> We hope our responses have satisfactorily addressed your key concerns. If anything remains unclear or requires further clarification, please do not hesitate to let us know, and we will address it promptly.
>
> We look forward to your feedback.
>
> Best regards,
>
> Authors of Submission 11733

---

> ### Author Response · Authors · 2024-11-30
> **Reminder - Discussion Stage Closing Soon - 1 Dec**
>
> Dear Reviewer WLby,
>
> Thank you for taking the time and effort to review our manuscript.
>
> We have carefully addressed all your comments and prepared detailed responses. Could you kindly review them at your earliest convenience?
>
> We hope our responses have satisfactorily addressed your key concerns. If anything remains unclear or requires further clarification, please do not hesitate to let us know, and we will address it promptly.
>
> We look forward to your feedback.
>
> Best regards,
>
> Authors of Submission 11733

---

> > ### Comment · Reviewer_WLby · 2024-12-01
> >
> > I thank the authors for their answers to my concerns and questions. I have decided to keep my score unchanged.

---

> > > ### Author Response · Authors · 2024-12-02
> > >
> > > Dear Reviewer WLby,
> > >
> > > Many thanks for your reply! We are not sure if your concerns are addressed well. If not, may we know your concerns after reading our responses?
> > >
> > > Best,
> > >
> > > Authors of Submission 11733

---

### Official Review · Reviewer_swsa · 2024-11-03

**Soundness:** 3
**Presentation:** 3
**Contribution:** 2
**Rating:** 6
**Confidence:** 4

**Summary:**

This paper proposes a method to learn representations that facilitates the detection of dependence. The authors instantiate their idea using variational estimators of mutual information and the Hilbert-Schmidt Independence Criterion. Finally, they provide theoretical and empirical evidence to demonstrate the effectiveness of their approach.

**Strengths:**

This paper is very well written and I was able to read it smoothly. The idea of minimizing the asymptotic power instead of the dependence metric itself is interesting. It is demonstrated in the experiments that their approach improves HSIC's power in high-dimensional problems.

**Weaknesses:**

A few concerns I have for this paper:

1. It seems to me that the authors didn't highlight their core contribution enough. The InfoNCE approach already exists in the literature and making an independence test out of it based on permutation is not very interesting; applying this trick to HSIC either. The interesting part to me is they propose to minimize the SNR -- a quantity that characterizes the statistical power -- instead of the test statistics themselves. However, this contribution is not supported in the experiments (or is not conveyed clearly). I would like to see a comparison between the InfoNCE (HSIC as well) that minimizes the lower bound (test statistic) versus the one that minimizes the SNR. If the latter shows significant improvement upon the former, I would then say the contribution is solid.

2. The authors didn't mention a closely related topic -- independence tests based on optimal transport (e.g., [1], [2], and references therein). In this line of work, the distance between the joint and the production of marginals is measured by the optimal transport distances.

[1] D. M. Cifarelli and E. Regazzini. On the centennial anniversary of Gini’s theory of statistical relations. Metron, 2017.

[2] L. Liu, S. Pal, Z. Harchaoui. Entropy Regularized Optimal Transport Independence Criterion. AISTATS, 2022.

**Questions:**

Questions:
1. Although using permutation tests is easy, it may be less effective and computationally costly. Is it possible to derive a limiting distribution for your test statistics? Also, the computational complexity of Phase Two in Algorithm 1 needs to be discussed.

Small comments:
1. Line 218. If $X \perp Y$ -> if $X \not\perp Y$.
2. Line 223. When HSIC = 0 the statistic is not mean zero.
3. Line 417 and 419. There is no $\lambda$ in $J_\lambda$.

---

> ### Author Response · Authors · 2024-11-22
>
> Thank you for your constructive feedback and your review of our paper.
>
> > It seems to me that the authors didn't highlight their core contribution enough. The InfoNCE approach already exists in the literature and making an independence test out of it based on permutation is not very interesting; applying this trick to HSIC either. The interesting part to me is they propose to minimize the SNR -- a quantity that characterizes the statistical power -- instead of the test statistics themselves. However, this contribution is not supported in the experiments (or is not conveyed clearly). I would like to see a comparison between the InfoNCE (HSIC as well) that minimizes the lower bound (test statistic) versus the one that minimizes the SNR. If the latter shows significant improvement upon the former, I would then say the contribution is solid.
>
> Thank you for your interest in our work. To summarize, we present two (related) approaches: one based on maximizing the asymptotic power of a HSIC test, and one based on maximizing variational mutual information bounds.
>
> Regarding the mutual information bound-based test, as in InfoNCE:
>
> We agree that constructing a permutation test from this bound is not a surprising technical contribution, but to our knowledge it has not been considered before in the literature, despite the ease of doing so; we think making this explicit for practitioners unfamiliar with one or the other set of tools is valuable in itself. We also think that the observation of the equivalence of essentially all of these bounds at test time, based on $\hat T$, is novel and interesting.
>
> As for maximizing the SNR:
>
> For HSIC, this is exactly what we do in all of our experiments: HSIC-D/Dx/O are all based on maximizing the $J$ objective, which is HSIC divided by its asymptotic standard deviation. As an ablation, we included tests where we just trained the HSIC statistic (rather than its SNR) in Figure 8 (Appendix C.5); while this approach is reasonably comparable to (but worse than) our method for HDGM-10, it is far far worse than maximizing the SNR for HDGM-30, -50, and RatInABox.
>
> For the $\hat T$-based tests corresponding to MI estimators, as you noticed, we did not originally explore maximizing the SNR in our experiments. In the last week since seeing the reviews, we have explored this idea in practice. Implementing this procedure yields a valid test, with empirical results in the new Appendix E.1. On Sinusoid, it performs in between InfoNCE and NWJ (which are all close); on RatInABox, it performs slightly worse than NWJ, which is notably worse than InfoNCE; on HDGM-4, its performance is poor; on HDGM-10, it performs abysmally, with trivial power. Throughout, it is consistently worse than our HSIC-based tests.
>
> Why did this happen? Appendix E.2.2 explores this question on RatInABox. Here, the $\hat T$-SNR test does obtain better SNRs than InfoNCE (by a factor of 2-3). In fact, the $\hat T$-SNR test follows the asymptotics quite closely, with observed powers very close to those predicted by our asymptotic formula $\Phi\left( \sqrt{m} \, \mathrm{SNR} - \frac{\sigma\_{\mathfrak{H}_0}}{ \sigma\_{\mathfrak{H}_1}} \Phi^{-1}(1-\alpha) \right)$.
>
> InfoNCE tests, though, do much, much better than predicted by these asymptotics. This is due to a strange phenomenon: as shown in Figure 15, the InfoNCE estimate and permutation-based rejection threshold are 99\% correlated across different observed test sets for the same problem. This strange behavior is totally outside of our asymptotic analysis, which assumes the threshold is approximately a constant independent of the particular test sample.
>
> This can also partially explain why $\hat T$-SNR tests are so much worse than HSIC ones. In this case, the ratio of standard deviations is approximately one; thus the power is about $\Phi\left( \sqrt{m} \, \mathrm{SNR} - 1.6 \right)$, for $\alpha = 0.05$. When $m = 2,000$, $\sqrt m$ is about $45$, and we obtain SNRs around $0.02$, giving a product around $0.9$. Thus subtracting $1.6$ from the probit of the power is highly significant until $m$ is much larger. By contrast, the asymptotic power of the HSIC test is $\Phi\left( \sqrt{m} J - \frac{r}{ \sigma\_{\mathfrak{H}_1} \sqrt{m}} \right)$; this additional $1/\sqrt{m}$ on the second term means that its relevance diminishes faster as $m$ grows, avoiding this weight which holds back $\hat T$-SNR tests.

---

> > ### Author Response · Authors · 2024-11-22
> >
> > > The authors didn't mention a closely related topic -- independence tests based on optimal transport (e.g., [1], [2], and references therein). In this line of work, the distance between the joint and the production of marginals is measured by the optimal transport distances.
> >
> > Thank you for bringing these papers to our attention. [2] in particular seems quite similar in form to HSIC, but with increased computational cost; they also only use fixed representations in their experiments, and so are not actually a direct comparison to our work. (Probably, work to optimize the choice of an optimal transport cost in [2] could be possible, and analogous to our paper.) We will be add these to the discussion of related work.
> >
> >
> >
> > > Although using permutation tests is easy, it may be less effective and computationally costly. Is it possible to derive a limiting distribution for your test statistics?
> >
> > Under the null hypothesis HSIC is degenerate, and so converges to a complex problem-dependent combination of chi-squared distributions. Some earlier work uses an entirely empirical, not-really-justified approximation with a Gamma distribution [Gretton et al., 2007]. It is possible to estimate this distribution using an eigendecomposition [Gretton et al., 2009], but that is in fact much more expensive than permutation for large sample sizes. The most promising approach is probably xHSIC [Shekhar et al., 2023], which gives a slightly less powerful test but with an asymptotically normal test statistic. These latter two approaches are both only asymptotically valid, while permutation testing gives exact finite-sample validity, and with a reasonable number of permutations it is only mildly conservative in practice, implying that its power is near-optimal for a given test statistic.
> >
> > We also want to emphasize that permutation is far from the most expensive part of running tests with our scheme. If we pre-compute the kernel matrices we just have to take a few hundred different quadratic forms of that matrix, which takes seconds even for fairly large test sizes. Moreover, we do not need to this it while optimizing the kernel, only once at the very end. In practice, optimizing the kernel is a much larger portion of our computation time.
> >
> >
> >
> > > Also, the computational complexity of Phase Two in Algorithm 1 needs to be discussed.
> >
> > Given that we use a quadratic time estimator for $\text{HSIC}_u$, the total complexity is of order $\mathcal O(pm^2)$, where $p$ is the total number of permutations. We will add this to the time complexity subsection.
> >
> >
> >
> > > small comments 1 and 3
> >
> > Thanks; we've fixed these.
> >
> > > Line 223. When HSIC = 0 the statistic is not mean zero.
> >
> > We use an unbiased estimator of HSIC, so the statistic's mean is in fact $m \cdot 0 = 0$.

---

> > > ### Comment · Reviewer_swsa · 2024-11-27
> > > **Response to Authors**
> > >
> > > I thank the authors for answering my questions. I will keep my score unchanged.

---

> > > > ### Author Response · Authors · 2024-11-30
> > > >
> > > > Dear Reviewer swsa,
> > > >
> > > > Many thanks for your reply! May we double-check that your concerns are addressed well? If so, many thanks for your support by maintaining your positive score. If not, please let us know your further concerns.
> > > >
> > > > Best regards,
> > > >
> > > > Authors of Submission 11733

---

### Official Review · Reviewer_WsaA · 2024-11-04

**Soundness:** 3
**Presentation:** 2
**Contribution:** 3
**Rating:** 5
**Confidence:** 3

**Summary:**

The paper discusses methods for improving independence tests between random variables, focusing on high-dimensional data where traditional tests struggle. It introduces two main approaches: using variational estimators of mutual information (MI), such as InfoNCE and NWJ, and leveraging the Hilbert-Schmidt Independence Criterion (HSIC). The authors demonstrate that variational MI-based tests can be closely related to kernel-based HSIC tests, with both methods aiming to detect dependencies more efficiently. They propose a new method that learns representations to maximize the power of HSIC tests, showing that this approach outperforms standard variational methods in detecting complex dependencies.

**Strengths:**

The paper is technically well presented, and the claims are well supported, theoretically and experimentally. It introduces novel approaches for learning representations that enhances the power of independence tests, particularly in high-dimensional settings, improving upon traditional methods. Additionally, the work demonstrates theoretical validity and empirical effectiveness of the proposed approaches, complemented by extensive evaluations that reinforce its contributions to the field.

**Weaknesses:**

- A notable issue is the lack of clarity regarding the aims and achievements of the paper. For instance, while it claims to present two approaches for independence testing, the conclusion suggests that it only studies one approach, leading to confusion about the overall contribution and focus of the work.
- The performance of HSIC-based tests is significantly influenced by the choice of kernel. While the paper proposes methods to optimize kernel selection, identifying the appropriate kernel for specific data types or distributions remains challenging and may not generalize well across diverse datasets.

- Although the proposed methods enhance sample efficiency compared to traditional tests, they still necessitate a considerable number of samples to effectively detect subtle dependencies in very high-dimensional data, which could limit their practicality in scenarios with restricted data availability.

- Additionally, the paper does not provide theoretical rates for the convergence of test statistics and lacks a discussion on the asymptotic validity of power in its main content.

- Furthermore, regarding Algorithm 1, the choice of $ n_{\text{perm}}$ is required; what are the best empirical strategies for selecting this parameter, and how does the choice impact the convergence rate?

**Questions:**

Please see the weakness section.

---

> ### Author Response · Authors · 2024-11-22
>
> Thank you for your constructive feedback and your review of our paper.
>
> > A notable issue is the lack of clarity regarding the aims and achievements of the paper. For instance, while it claims to present two approaches for independence testing, the conclusion suggests that it only studies one approach, leading to confusion about the overall contribution and focus of the work.
>
> Thank you for bringing this to our attention. We indeed present two related approaches to independence testing: one based on maximizing the asymptotic power of a HSIC test, and one family based on maximizing variational mutual information bounds. We will be sure to make both contributions more explicit in the conclusion.
>
> > The performance of HSIC-based tests is significantly influenced by the choice of kernel. While the paper proposes methods to optimize kernel selection, identifying the appropriate kernel for specific data types or distributions remains challenging and may not generalize well across diverse datasets.
>
> It is true that the choice of kernel significantly impacts the test performance. Using a fixed test may not work well on any given problem, which is indeed the entire purpose of our paper. Using a particular kernel architecture to optimize for a given problem may not help, if the kernel architecture is inappropriate; this is exactly analogous to the same concern in supervised learning, however, and we can choose between different options of kernel architecture in the same way that we choose between different options of classifier architecture when training a classifier. Our experiments confirm that a general MLP-type architecture feeding into a Gaussian kernel gives a good tradeoff between expressivity and ability to learn in the class; note in particular that HSIC-D, using a deep kernel, always outperforms HSIC-O, choosing the bandwidth of a Gaussian kernel.
>
> > Although the proposed methods enhance sample efficiency compared to traditional tests, they still necessitate a considerable number of samples to effectively detect subtle dependencies in very high-dimensional data, which could limit their practicality in scenarios with restricted data availability.
>
> It is true that learning a kernel requires more data than just using an off-the-shelf kernel; this is an instance of the tradeoff mentioned in the previous answer. For very small sample sizes, we should choose a simpler kernel architecture; for very small sizes, not optimizing will be better than trying to optimize. The experiments on Sinusoid and RatInABox show that even for fairly complex forms of dependence with fairly general kernel architectures, the total number of training samples needed to identify a good kernel need not be exorbitant (<2,000 here).
>
> As our experiments show, our method is better than many alternatives for several problems across a wide range of total available samples. For instance, although HSICAgg does not need a separate training set, our train+test split method typically outperforms it.
>
> > Additionally, the paper does not provide theoretical rates for the convergence of test statistics and lacks a discussion on the asymptotic validity of power in its main content.
>
> We briefly mentioned the already-known asymptotics/convergence rate for these statistics around line 220; we used these results to derive our HSIC-D method in line 308. Detailed theorems regarding HSIC asymptotics are given in Appendix A, as a direct result of the asymptotic normality of U-statistics [e.g. Serfling 1980, Theorem B, p.193]. For the revision, we will try to bring more details around this into the main body to provide more context for the reader; thanks for bringing this to our attention
>
> > Furthermore, regarding Algorithm 1, the choice of $n_\text{perm}$ is required; what are the best empirical strategies for selecting this parameter, and how does the choice impact the convergence rate?
>
> $n_\text{perm}$ is the number of shuffles used in the permutation test, and does not affect the convergence rate; it is totally independent of the choice of test statistic/kernel leading into the permutation test, and the choice does not affect the training procedure. As long as we include the original alignment in the permutation procedure, the permutation test is guaranteed to be finite-sample valid [Hemerik & Goeman 2018, Theorem 2]. The conservatism of the $p$-value is determined by the number of permutations; a one-permutation test is extremely conservative, while using all $n!$ permutations when $n \alpha$ is also an integer would yield an exact $p$-value. Larger $n_\text{perm}$ will give a slightly less conservative test, but the difference beyond using a few hundred permutations is minimal for typical values of $\alpha$. We also emphasize that we do not need to permute during training, only once at the very end, and so using a few hundred permutations (each of which only requires taking a quadratic form of the kernel matrix) is not a substantial portion of our runtime.

---

> ### Author Response · Authors · 2024-11-28
> **Reminder - Discussion Stage Closing Soon - 28 November**
>
> Dear Reviewer WsaA,
>
> Thank you for taking the time and effort to review our manuscript.
>
> We have carefully addressed all your comments and prepared detailed responses. Could you kindly review them at your earliest convenience?
>
> We hope our responses have satisfactorily addressed your key concerns. If anything remains unclear or requires further clarification, please do not hesitate to let us know, and we will address it promptly.
>
> We look forward to your feedback.
>
> Best regards,
>
> Authors of Submission 11733

---

> ### Author Response · Authors · 2024-11-30
> **Reminder - Discussion Stage Closing Soon - 1 Dec**
>
> Dear Reviewer WsaA,
>
> Thank you for taking the time and effort to review our manuscript.
>
> We have carefully addressed all your comments and prepared detailed responses. Could you kindly review them at your earliest convenience?
>
> We hope our responses have satisfactorily addressed your key concerns. If anything remains unclear or requires further clarification, please do not hesitate to let us know, and we will address it promptly.
>
> We look forward to your feedback.
>
> Best regards,
>
> Authors of Submission 11733

---

> > ### Comment · Reviewer_WsaA · 2024-12-01
> >
> > Thank you for your time and effort. I will thoroughly the answers and take it into consideration before finalizing my score at the end of the review period.

---

> > > ### Comment · Reviewer_WsaA · 2024-12-02
> > >
> > > Dear Authors,
> > >
> > > Thank you for your thoughtful responses to my concerns and questions. After carefully reviewing your clarifications, I have decided to maintain my original score.
> > >
> > > I appreciate the effort you have put into addressing the issues.
> > >
> > > Best regards,
> > >
> > > Reviewer WsaA

---

> > > > ### Author Response · Authors · 2024-12-02
> > > >
> > > > Dear Reviewer WsaA,
> > > >
> > > > Many thanks for your reply! We are not sure if your concerns are addressed well. If not, may we know your concerns after reading our responses?
> > > >
> > > > Best,
> > > >
> > > > Authors of Submission 11733

---

### Meta-Review · Area_Chair_zsHx · 2024-12-18

**Metareview:**

This work explores two approaches to enhance independence testing between random variables, crucial for machine learning and statistics. The first approach uses variational estimators of mutual information, like InfoNCE and NWJ, to construct powerful finite-sample valid tests, while the second is based on the Hilbert-Schmidt Independence Criterion (HSIC).  The paper demonstrates learning a variational bound in the former is closely related to learning kernels in the latter.  Empirical results show the benefit of learning representations for the HSIC based method in detecting complex structured dependencies.

The paper is well written and has interesting theoretical and practical results.  The empirical evaluation on real world datasets is still quite limited, and more analysis is needed on the sample complexity related to kernel learning.

**Additional Comments On Reviewer Discussion:**

The rebuttal has been noted by the reviewers and have been taken into account by the AC in the recommendation of acceptance/rejection.

---

### Decision · Program_Chairs · 2025-01-22

Reject